# Unsupervised Image Denoising with Score Function

**Yutong Xie**
Peking University
`yutongxie_research@163.com`

**Mingze Yuan**
Peking University
`mzyuan@pku.edu.cn`

**Bin Dong**
Peking University
`dongbin@bicmr.pku.edu.cn`

**Quanzheng Li**
Massachusetts General Hospital and Harvard Medical School
`li.quanzheng@mgh.harvard.edu`

## Abstract

Though achieving excellent performance in some cases, current unsupervised learning methods for single image denoising usually have constraints in applications. In this paper, we propose a new approach which is more general and applicable to complicated noise models. Utilizing the property of score function, the gradient of logarithmic probability, we define a solving system for denoising. Once the score function of noisy images has been estimated, the denoised result can be obtained through the solving system. Our approach can be applied to multiple noise models, such as the mixture of multiplicative and additive noise combined with structured correlation. Experimental results show that our method is comparable when the noise model is simple, and has good performance in complicated cases where other methods are not applicable or perform poorly.

## 1 Introduction

Image denoising [4, 20, 26] has been studied for many years. Suppose $x$ is a clean image, $y$ is a noisy image of $x$, and $p(y \mid x)$ is the noise model. Supervised learning methods try to train a model representing a mapping from $y$ to $x$. Due to the difficulty of collecting paired clean and noisy images in practice, methods in unsupervised learning manner are focus of research. Noise2Noise [12] is the first to use pairs of two different noisy images constructed from the same clean image to train a denoising model. Strictly speaking, Noise2Noise is not an unsupervised learning method. Collecting noisy pairs is still difficult. Despite all this, many other methods [2, 10, 11, 24, 25] are inspired from Noise2Noise or borrow the idea behind of it. These methods achieve good performance in some simple noise models.

The main drawback of current methods is the constraint of application. Once the noise model is complicated, they either are not applicable or have poor performance. Noisier2Noise [16] can only handle additive noise and pepper noise. Recorrupted-to-Recorrupted [17] is limited to Gaussian noise. Neighbor2Neighbor [7] requires that the noise model is pixel-wise independent and unbiased (*i.e.* $\mathbb{E}[y \mid x] = x$). Noise2Score [9] applies Tweedie's Formula to image denoising and provides a unified framework for those noise models that follow exponential family distributions. However, the practical noise model will be more complicated. It may contain both multiplicative and additive noise, even combining with structural correlation.

In this paper, we propose a new unified approach to handle more noise models. The key of our approach is a theoretical property about score function, $\nabla_y \log p(y)$, which is shown in Proposition 3.1. This property indicates that the score function of $y$ is the average of the score function of $y \mid x$ under the posterior distribution $p(x \mid y)$. Based on it, we define a system that the score function of $y$ equals to the score function of $y \mid x$, which turns out to be an equation about $x$ since $y$ is known. We

37th Conference on Neural Information Processing Systems (NeurIPS 2023).

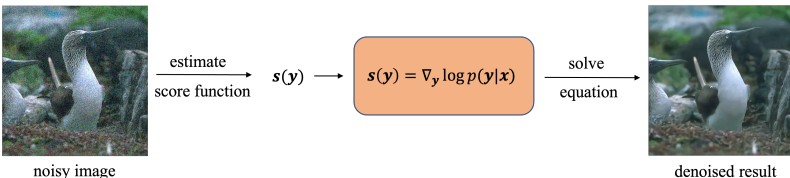

noisy image       denoised result

Figure 1: The overall flow of our approach. The first step is to estimate the score function $s(y)$ and the second step is to solve an equation.

discover that the solution of this equation can be regarded as the denoised result of $y$. Implementing our approach practically contains two steps: the first is to estimate the score function of $y$ and the second is to solve the system defined above according to the specific noise model. The overall flow is shown in Figure 1. All the model training is the estimation of score function, which is represented by a neural network. We adapt the amortized residual denoising autoencoder (AR-DAE) [13] to figure out it. More details can be seen in Section 3.3.

Our approach is so powerful that as long as the score function of $y \mid x$ is known and the system is solvable, any kind of noise model is applicable. It means our approach is even able to solve sophisticated noise models such as mixture noise. Another advantage of our approach is that regardless of noise models, the training process of the score function neural network is identical. Therefore, once the assumption of the noise model does not hold or parameters of noise are corrected, we only change the equation system to be solved and resolve it without training the model again. In summary, our main contribution are: (1) We propose a general unsupervised approach for image denoising, which is based on the score function. (2) Experimental results show that our approach is competitive for simple noise models and achieves excellent performance for complicated noise models where other unsupervised methods is invalid.

## 2 Related Works

Here we briefly review the existing deep learning methods for image denoising. When the pairs of clean images and noisy images are available, the supervised training [26] is to minimize $\mathbb{E}_{x,y}\left[d\left(x, f(y; \theta)\right)\right]$, where $f(\cdot; \theta)$ is a neural network and $d(\cdot, \cdot)$ is a distance metric. Though supervised learning has great performance, the difficult acquisition for training data hampers its application in practice.

To avoid the issues of acquisition for paired clean and noisy images, unsupervised[1] approaches are proposed to use noisy image pairs to learn image denoising, which is started from Noise2Noise (N2N) [12]. While N2N can achieve comparable results with supervised methods, collecting noisy image pairs from real world is still intracTable Motivated by N2N, the followed-up works try to learn image denoising with individual noisy images. Mask-based unsupervised approaches [2] design mask schemes for denoising on individual noisy images and then they train the network to predict the masked pixels according to noisy pixels in the input receptive field. Noise2Void [10] also proposes the blind-spot network (BSN) to avoid learning the identity function. Noisier2Noise [16], Noisy-As-Clean [25] and Recorrupted-to-Recorrupted [17] require a single noisy realization of each training sample and a statistical model of the noise distribution. Noisier2Noise first generates a synthetic noisy sample from the statistical noise model, adds it to the already noisy image, and asks the network to predict the original noisy image from the doubly noisy image. Besides the blind-spot network design, Self2Self [19] is proposed on blind denoising to generate paired data from a single noisy image by applying Bernoulli dropout. Recently, Neighbor2Neighbor [7] proposes to create subsampled paired images based on the pixel-wise independent noise assumption and then a denoising network is trained on generated pairs, with additional regularization loss for better performance. Noise2Score [9] is another type of unsupervised methods that can deal with any noise model which follows an exponential family distribution. It utilize Tweedie's Formula [21, 6] and the estimation of score function to denoise noisy images.

---

[1]Sometimes self-supervised is used.

# 3 Method

In this section, we provide a more detailed description of the proposed approach. The organization is as follows: In Section 3.1 we review Noise2Score, introduce the basic theory and present the relationship between our method and Noise2Score. After that, we derive the specific algorithms for different examples of noise models in Section 3.2. Finally, we describe the method for estimating the score function $\nabla_{\boldsymbol{y}} \log p(\boldsymbol{y})$ in Section 3.3. All the proofs are in Supplementary Material.

## 3.1 Basic Theory

### 3.1.1 Noise2Score

Suppose the noise model follows some exponential family distribution and has the following form:

$$p(\boldsymbol{y} \mid \boldsymbol{x}) = b(\boldsymbol{y}) \exp \left\{ H(\boldsymbol{x})^{\top} T(\boldsymbol{y}) - a(\boldsymbol{x}) \right\}, \tag{1}$$

where $T(\boldsymbol{y})$ and $(\boldsymbol{x})$ have the same dimensions, and $a(\boldsymbol{x})$ and $b(\boldsymbol{y})$ are both scalar functions. One of the properties of an exponential family distribution is that:

$$H'(\boldsymbol{x})^{\top} \mathbb{E}\left[T(\boldsymbol{y}) \mid \boldsymbol{x}\right] = a'(\boldsymbol{x}). \tag{2}$$

According to Bayesian Formula, we have the following derivation:

$$p(\boldsymbol{x} \mid \boldsymbol{y}) = \frac{p(\boldsymbol{x}) p(\boldsymbol{y} \mid \boldsymbol{x})}{p(\boldsymbol{y})} = p(\boldsymbol{x}) e^{-a(\boldsymbol{x})} \exp \left( T(\boldsymbol{y})^{\top} H(\boldsymbol{x}) + \log \frac{b(\boldsymbol{y})}{p(\boldsymbol{y})} \right). \tag{3}$$

Therefore, $p(\boldsymbol{x} \mid \boldsymbol{y})$ is also an exponential family distribution. According to Eq. 2, we can derive a conclusion about $\nabla_{\boldsymbol{y}} \log p(\boldsymbol{y})$ (the score function of $\boldsymbol{y}$), Tweedie's Formula, that

$$\nabla_{\boldsymbol{y}} \log p(\boldsymbol{y}) = \nabla_{\boldsymbol{y}} \log b(\boldsymbol{y}) + T'(\boldsymbol{y})^{\top} \mathbb{E}\left[H(\boldsymbol{x}) \mid \boldsymbol{y}\right]. \tag{4}$$

In Noise2Score, the denoised result $\hat{\boldsymbol{x}}$ of $\boldsymbol{y}$ is the solution of following equation:

$$\nabla_{\boldsymbol{y}} \log p(\boldsymbol{y}) = \nabla_{\boldsymbol{y}} \log b(\boldsymbol{y}) + T'(\boldsymbol{y})^{\top} H(\hat{\boldsymbol{x}}). \tag{5}$$

Since Noise2Score is derived from Tweedie's Formula, it is limited to the noise models of exponential family distributions.

### 3.1.2 Proposed Approach

In this work, the proposed approach generalized Noise2Score to non-exponential family distributions. We begin with the following proposition.

**Proposition 3.1** *Let $\boldsymbol{y} \sim \mathrm{y}$ and $\boldsymbol{x} \sim \mathrm{x}$ where $\mathrm{x}$ and $\mathrm{y}$ are two random variable, then the equation below holds:*

$$\nabla_{\boldsymbol{y}} \log p(\boldsymbol{y}) = \int p(\boldsymbol{x} \mid \boldsymbol{y}) \nabla_{\boldsymbol{y}} \log p(\boldsymbol{y} \mid \boldsymbol{x}) \, \mathrm{d}\boldsymbol{x} \tag{6}$$

The proof of Proposition 3.1 is in Supplementary Material. In the task of image denoising, suppose $\boldsymbol{y} \in \mathbb{R}^{d}$ is the noisy image and $\boldsymbol{x} \in \mathbb{R}^{d}$ is the clean image where $d$ represents the number of image pixels. $p(\boldsymbol{x} \mid \boldsymbol{y})$ represents the posterior distribution and $p(\boldsymbol{y} \mid \boldsymbol{x})$ denotes the noise model. When $p(\boldsymbol{y} \mid \boldsymbol{x})$ follows some exponential family distribution, it is easy to verify that the right part of Eq. 6 equals to the right part of Tweedie's Formula (Eq. 4). Therefore, the equation (Eq. 5) to be solved in Noise2Score in essence can be written as:

$$\nabla_{\boldsymbol{y}} \log p(\boldsymbol{y}) = \nabla_{\boldsymbol{y}} \log p(\boldsymbol{y} \mid \hat{\boldsymbol{x}}). \tag{7}$$

We use $\hat{\boldsymbol{x}}$ rather than $\boldsymbol{x}$ to indicate that $\hat{\boldsymbol{x}}$ is the unknown variable to be solved in the equation. Hence, we propose a new unsupervised denoising method that given noisy image $\boldsymbol{y}$, **the denoised result can be obtained by solving Eq. 7**. When the noise model follows some exponential family distribution, our method is the same as Noise2Score. The specific conclusions of Gaussian, Gamma and Poisson noise can be seen in Supplementary Material. However, different from Noise2Score, our approach is applicable to some complex non-exponential family distributions as long as the score function of $\boldsymbol{y}$ and noise model $p(\boldsymbol{y} \mid \boldsymbol{x})$ are known. For convenience, let $\boldsymbol{s}(\boldsymbol{y}) = \nabla_{\boldsymbol{y}} \log p(\boldsymbol{y})$ and $\boldsymbol{f}(\boldsymbol{x}, \boldsymbol{y}) = \nabla_{\boldsymbol{y}} \log p(\boldsymbol{y} \mid \boldsymbol{x})$. Our approach is consist of two steps as shown in Figure 1. The general algorithm is illustrated in Algorithm 1.

The following theorem explains why the solution of Eq. 7 is a good denoised result of $\boldsymbol{y}$.

---

**Algorithm 1** The general denoising process

---
**Input:** noisy image $\boldsymbol{y}$ and $\boldsymbol{f}(\boldsymbol{x}, \boldsymbol{y})$.
**Output:** $\hat{\boldsymbol{x}}$, the solution of Eq. 7.
 1: Estimate $\boldsymbol{s}(\boldsymbol{y})$.
 2: Solve $\boldsymbol{s}(\boldsymbol{y}) = \boldsymbol{f}(\boldsymbol{x}, \boldsymbol{y})$.

---

**Theorem 3.1** *Given $\boldsymbol{y}$, suppose the Hessian matrix $\boldsymbol{f}(\boldsymbol{x}, \boldsymbol{y})$ is bounded, $\boldsymbol{f}(\boldsymbol{x}, \boldsymbol{y})$ is invertible with respect to $\boldsymbol{x}$, and its inverse function is Lipschitz continuous. Denote $\hat{\boldsymbol{x}}$ as the solution of Eq. 7, then,*

$$\left\| \hat{\boldsymbol{x}} - \mathbb{E}_{\boldsymbol{x}|\boldsymbol{y}}[\boldsymbol{x}] \right\|_2 \leq C \mathrm{Tr}\left( \mathrm{Cov}\left[ \boldsymbol{x} \mid \boldsymbol{y} \right] \right) + o\left( \mathrm{Tr}\left( \mathrm{Cov}\left[ \boldsymbol{x} \mid \boldsymbol{y} \right] \right) \right), \tag{8}$$

*where $\mathrm{Cov}\left[ \boldsymbol{x} \mid \boldsymbol{y} \right]$ is the covariance matrix and $C$ is a constant.*

The proof of Theorem 3.1 is in Supplementary Material. Theorem 3.1 shows the lower noise level, the smaller the error between $\hat{\boldsymbol{x}}$ and $\mathbb{E}_{\boldsymbol{x}|\boldsymbol{y}}[\boldsymbol{x}]$, which guarantees the effectiveness of proposed method.

Next, in Section 3.2 we will derive the specific algorithm for solving $\boldsymbol{x}$ in Eq. 7 under different noise models, including additive Gaussian noise in Section 3.2.1, multiplicative noise in Section 3.2.2 and mixture noise in Section 3.2.3. Estimating score function $\boldsymbol{s}(\boldsymbol{y})$ will be introduced in Section 3.3.

### 3.2 Application Examples

#### 3.2.1 Additive Gaussian Noise

Suppose the noise model is $\boldsymbol{y} = \boldsymbol{x} + \boldsymbol{\epsilon}$ where $\boldsymbol{\epsilon}$ follows a multi-variable Gaussian distribution with mean of $\boldsymbol{0}$ and covariance matrix of $\boldsymbol{\Sigma}$ denoted by $\mathcal{N}(\boldsymbol{0}, \boldsymbol{\Sigma})$, *i.e.*

$$p(\boldsymbol{y} \mid \boldsymbol{x}) = \frac{\exp\left\{ -\frac{1}{2}(\boldsymbol{y} - \boldsymbol{x})^\top \boldsymbol{\Sigma}^{-1}(\boldsymbol{y} - \boldsymbol{x}) \right\}}{\sqrt{2\pi}^d |\boldsymbol{\Sigma}|^{\frac{1}{2}}} \tag{9}$$

Then, we derive that $\boldsymbol{f}(\boldsymbol{x}, \boldsymbol{y}) = \nabla_{\boldsymbol{y}} \log p(\boldsymbol{y} \mid \boldsymbol{x}) = -\boldsymbol{\Sigma}^{-1}(\boldsymbol{y} - \boldsymbol{x})$. We consider the following four kinds of $\boldsymbol{\Sigma}$: (1) $\boldsymbol{\Sigma} = \sigma^2 \boldsymbol{I}$; (2) $\boldsymbol{\Sigma} = \sigma^2 \boldsymbol{A}^\top \boldsymbol{A}$; (3) $\boldsymbol{\Sigma} = \boldsymbol{\Sigma}(\boldsymbol{x}) = \mathrm{diag}\left( a\boldsymbol{x} + b\boldsymbol{1} \right)^2$; (4) $\boldsymbol{\Sigma} = \boldsymbol{\Sigma}(\boldsymbol{x}) = \boldsymbol{A}^\top \mathrm{diag}\left( a\boldsymbol{x} + b\boldsymbol{1} \right)^2 \boldsymbol{A}$. In the second and fourth cases, $\boldsymbol{A}$ usually represents a convolution transform which describes the correlation between adjacent pixels.

For the first and second cases, $\boldsymbol{\Sigma}$ is a constant matrix. By solving Eq. 7, we have

$$\hat{\boldsymbol{x}} = \boldsymbol{\Sigma} \boldsymbol{s}(\boldsymbol{y}) + \boldsymbol{y}. \tag{10}$$

While in the third and fourth cases, $\boldsymbol{\Sigma}$ is related to $\boldsymbol{x}$ and Eq. 10 is a fixed point equation. Therefore, we use an iterative trick to solve it as shown in Algorithm 2.

---

**Algorithm 2** An iterative trick to solve $\boldsymbol{x} = \boldsymbol{\Sigma}(\boldsymbol{x}) \boldsymbol{s}(\boldsymbol{y}) + \boldsymbol{y}$

---
**Input:** noisy image $\boldsymbol{y}$, $\boldsymbol{s}(\boldsymbol{y})$, the parameters of $\boldsymbol{\Sigma}(\cdot)$ and the number of iterations $n$.
**Output:** $\hat{\boldsymbol{x}}$, the solution.
 1: Initial value of $\hat{\boldsymbol{x}}$ is set as $\boldsymbol{y}$.
 2: **for** $i = 1, ..., n$ **do**
 3: $\quad \hat{\boldsymbol{x}} \leftarrow \boldsymbol{\Sigma}(\hat{\boldsymbol{x}}) \boldsymbol{s}(\boldsymbol{y}) + \boldsymbol{y}$.
 4: **end for**

---

#### 3.2.2 Multiplicative Noise

Firstly, we discuss three types of multiplicative noise model, Gamma, Poisson and Rayleigh noise. Then, we consider the situation where the convolution transform $\boldsymbol{A}$ exists.

**Gamma Noise** is constructed from Gamma distribution, $\mathcal{G}(\alpha, \beta)$: $p(x; \alpha, \beta) = \frac{\beta^\alpha}{\Gamma(\alpha)} x^{\alpha-1} e^{-\beta x}$, and is defined as $\boldsymbol{y} = \boldsymbol{\eta} \odot \boldsymbol{x}$, where $\eta_i \sim \mathcal{G}(\alpha, \alpha)$, $\alpha > 1$, and $\odot$ means component-wise multiplication, *i.e.*

$$p(\boldsymbol{y} \mid \boldsymbol{x}) = \prod_{i=1}^{d} \frac{\alpha^\alpha}{\Gamma(\alpha)} \left( \frac{y_i}{x_i} \right)^{\alpha-1} \exp\left\{ -\frac{\alpha y_i}{x_i} \right\} \cdot \frac{1}{x_i}. \tag{11}$$

Then, we derive that $\boldsymbol{f}\left(\boldsymbol{x}, \boldsymbol{y}\right) = \nabla_{\boldsymbol{y}} \log p\left(\boldsymbol{y} \mid \boldsymbol{x}\right) = \frac{\alpha-1}{\boldsymbol{y}} - \frac{\alpha}{\boldsymbol{x}}$. Here, the division is component-wise division. In the residual part of this paper, we neglect such annotation if it is not ambiguous. By solving Eq. 7, we have

$$\hat{\boldsymbol{x}} = \frac{\alpha \boldsymbol{y}}{\alpha - 1 - \boldsymbol{y} \odot \boldsymbol{s}\left(\boldsymbol{y}\right)}. \tag{12}$$

**Poisson Noise** is constructed from Poisson distribution, $\mathcal{P}\left(\lambda\right)$: $\Pr(x=k) = \frac{\lambda^k}{k!} e^{-\lambda}, k = 0, 1, \cdots$, and is defined as $\boldsymbol{y} = \frac{1}{\lambda} \boldsymbol{\eta}, \eta_i \sim \mathcal{P}\left(\lambda x_i\right), \lambda > 0$, *i.e.*

$$\Pr\left(\boldsymbol{y} \mid \boldsymbol{x}\right) = \prod_{i=1}^{d} \frac{\left(\lambda x_i\right)^{\lambda_i y_i}}{\left(\lambda y_j\right)!} e^{-\lambda x_i}. \tag{13}$$

Then, we derive that $\boldsymbol{f}\left(\boldsymbol{x}, \boldsymbol{y}\right) = \nabla_{\boldsymbol{y}} \log \Pr\left(\boldsymbol{y} \mid \boldsymbol{x}\right) = \lambda \log\left(\lambda \boldsymbol{x}\right) - \lambda \log\left(\lambda \boldsymbol{y} + \frac{1}{2}\right)$. By solving Eq. 7, we have

$$\hat{\boldsymbol{x}} = \left(\boldsymbol{y} + \frac{1}{2\lambda}\right) \odot \exp\left\{\frac{\boldsymbol{s}\left(\boldsymbol{y}\right)}{\lambda}\right\}, \tag{14}$$

**Rayleigh Noise** is constructed from Rayleigh distribution, $\mathcal{R}\left(\sigma\right)$: $p(x; \sigma) = \frac{x}{\sigma^2} \exp\left\{-\frac{x^2}{2\sigma^2}\right\}$, and is defined as $\boldsymbol{y} = \left(\boldsymbol{\eta} + 1\right) \odot \boldsymbol{x}, \eta_i \sim \mathcal{R}\left(\sigma\right), \sigma > 0$, *i.e.*

$$p\left(\boldsymbol{y} \mid \boldsymbol{x}\right) = \prod_{i=1}^{d} \frac{1}{x_i} \frac{y_i - x_i}{x_i \sigma^2} \exp\left\{-\frac{\left(y_i - x_i\right)^2}{2x_i^2 \sigma^2}\right\}. \tag{15}$$

Then, we derive that $\boldsymbol{f}\left(\boldsymbol{x}, \boldsymbol{y}\right) = \nabla_{\boldsymbol{y}} \log p\left(\boldsymbol{y} \mid \boldsymbol{x}\right) = \frac{1}{\boldsymbol{y}-\boldsymbol{x}} - \frac{\boldsymbol{y}-\boldsymbol{x}}{\sigma^2 \boldsymbol{x}^2}$. Solving Eq. 7 directly is not easy. Here we provide an iterative algorithm to solve it. It is illustrated in Algorithm 3.

---

**Algorithm 3** An iterative method to solve Eq. 7 in the case of Rayleigh noise

---

**Input:** noisy image $\boldsymbol{y}$, $\boldsymbol{s}\left(\boldsymbol{y}\right)$, the parameter of Rayleigh noise $\sigma$ and the number of iterations $n$.
**Output:** $\hat{\boldsymbol{x}}$, the solution of Eq. 7.
  1: Initial value of $\hat{\boldsymbol{x}}$ is set as $\boldsymbol{y}$.
  2: **for** $i = 1, ..., n$ **do**
  3:    Compute $\boldsymbol{b} = \sigma^2 \boldsymbol{s}\left(\boldsymbol{y}\right) \odot \hat{\boldsymbol{x}}$.
  4:    Compute $\boldsymbol{t} = \frac{1}{2}\left(-\boldsymbol{b} + \sqrt{\boldsymbol{b} \odot \boldsymbol{b} + 4\sigma^2 \boldsymbol{1}}\right)$.
  5:    $\hat{\boldsymbol{x}} \leftarrow \frac{\boldsymbol{y}}{\boldsymbol{t}+1}$.
  6: **end for**

---

Now, we consider the situation where the convolution transform $\boldsymbol{A}$ exists. Suppose the noise model is represented by $\boldsymbol{y} = \boldsymbol{A}\boldsymbol{z}, \boldsymbol{z} = N\left(\boldsymbol{x}\right)$, where $N\left(\boldsymbol{x}\right)$ can be any multiplicative noise model discussed above. Then, we have $\nabla_{\boldsymbol{y}} \log p_{\boldsymbol{y}}\left(\boldsymbol{y} \mid \boldsymbol{x}\right) = \boldsymbol{A}^{-1, \top} \nabla_{\boldsymbol{z}} \log p_{\boldsymbol{z}}\left(\boldsymbol{A}^{-1}\boldsymbol{y} \mid \boldsymbol{x}\right)$. To avoid confusion, we use subscripts to distinguish different distribution. Therefore, we can apply Algorithm 4 to solve Eq. 7, which is shown as follows.

---

**Algorithm 4** The general framework to solve Eq. 7 for correlated multiplicative noise model

---

**Input:** $\boldsymbol{y}$, $\boldsymbol{s}\left(\boldsymbol{y}\right)$, $\boldsymbol{A}$ and $\tilde{\boldsymbol{f}}\left(\boldsymbol{x}, \boldsymbol{z}\right) = \nabla_{\boldsymbol{z}} \log p_{\boldsymbol{z}}\left(\boldsymbol{z} \mid \boldsymbol{x}\right)$.
**Output:** $\hat{\boldsymbol{x}}$, the solution of Eq. 7.
  1: Computing $\tilde{\boldsymbol{s}} = \boldsymbol{A}^{\top} \boldsymbol{s}\left(\boldsymbol{y}\right)$.
  2: Computing $\boldsymbol{z} = \boldsymbol{A}^{-1}\boldsymbol{y}$.
  3: Solve $\tilde{\boldsymbol{s}} = \tilde{\boldsymbol{f}}\left(\boldsymbol{x}, \boldsymbol{z}\right)$ by the corresponding Algorithm

---

### 3.2.3 Mixture Noise

In this paper, the mixture noise model is composed of a multiplicative noise and an additive Gaussian noise. We denote it as $\boldsymbol{y} = \boldsymbol{z} + \boldsymbol{\epsilon}, \boldsymbol{\epsilon} \sim \mathcal{N}\left(0, \sigma^2 \boldsymbol{I}\right), \boldsymbol{z} = \boldsymbol{A}N\left(\boldsymbol{x}\right)$, where $N\left(\boldsymbol{x}\right)$ is any multiplicative

noise model that can be solved by our approach and $\boldsymbol{A}$ is either a convolution transform or identity matrix. It is easy to derive that $p_{\mathbf{y}}\left(\boldsymbol{y} \mid \boldsymbol{x}\right) = \int p_{\mathbf{y}}\left(\boldsymbol{y} \mid \boldsymbol{z}\right) p_{\mathbf{z}}\left(\boldsymbol{z} \mid \boldsymbol{x}\right) \mathrm{d}\boldsymbol{z}$. Generally speaking, $p_{\mathbf{y}}\left(\boldsymbol{y} \mid \boldsymbol{x}\right)$ has not an explicit analytical form. In this paper, we assume that the additive Gaussian noise is far smaller than the multiplicative noise. Thus, we utilize Taylor expansion to approximate $p_{\mathbf{y}}\left(\boldsymbol{y} \mid \boldsymbol{x}\right)$. We have the following conclusion:

$$p_{\mathbf{y}}\left(\boldsymbol{y} \mid \boldsymbol{x}\right) \approx p_{\mathbf{z}}\left(\bar{\boldsymbol{z}} \mid \mathbf{x}\right) + \nabla_{\boldsymbol{z}} p_{\mathbf{z}}\left(\bar{\boldsymbol{z}} \mid \boldsymbol{x}\right)^{T}\left(\boldsymbol{y} - \bar{\boldsymbol{z}}\right), \tag{16}$$

where $\bar{\boldsymbol{z}} = \mathbb{E}\left[\boldsymbol{z} \mid \boldsymbol{y}\right]$. Then, we can further derive that

$$\nabla_{\boldsymbol{y}} \log p_{\mathbf{y}}\left(\boldsymbol{y} \mid \boldsymbol{x}\right) \approx \nabla_{\boldsymbol{z}} \log p_{\mathbf{z}}\left(\bar{\boldsymbol{z}} \mid \boldsymbol{x}\right). \tag{17}$$

The full and rigorous derivations of Eq. 16 and Eq. 17 are in Supplementary Material. Applying Eq. 10 in Section 3.2.1, we have $\bar{\boldsymbol{z}} = \boldsymbol{y} + \sigma^2 \boldsymbol{s}\left(\boldsymbol{y}\right)$. Thus, the equation to be solve is $\boldsymbol{s}\left(\boldsymbol{y}\right) = \boldsymbol{f}\left(\boldsymbol{x}, \bar{\boldsymbol{z}}\right)$. The full denoising process is illustrated in Algorithm 5.

---

**Algorithm 5** The full denoising process for mixture noise $\boldsymbol{y} = \boldsymbol{z} + \boldsymbol{\epsilon}$

---

**Input:** $\boldsymbol{y}, \boldsymbol{s}\left(\boldsymbol{y}\right)$.
**Output:** $\hat{\boldsymbol{x}}$, the solution of $\boldsymbol{s}\left(\boldsymbol{y}\right) = \boldsymbol{f}\left(\boldsymbol{x}, \bar{\boldsymbol{z}}\right)$.
  1: Computing $\bar{\boldsymbol{z}} = \boldsymbol{y} + \sigma^2 \boldsymbol{s}\left(\boldsymbol{y}\right)$.
  2: Solve $\boldsymbol{s}\left(\boldsymbol{y}\right) = \boldsymbol{f}\left(\boldsymbol{x}, \bar{\boldsymbol{z}}\right)$ through the corresponding algorithm discussed in Section 3.2.2.

---

### 3.3 Estimation of Score Function

So far, we assume that the score function of $\boldsymbol{y}$, $\boldsymbol{s}\left(\boldsymbol{y}\right)$, is known. However, it is usually unknown and should be estimated from the dataset of noisy images $\{\boldsymbol{y}\}$. We use the same method, the amortized residual Denoising Auto Encoder (AR-DAE) [13] discussed in Noise2Score. Suppose $\boldsymbol{s}\left(\cdot; \theta\right)$ is a neural network used to represent the score function of $\boldsymbol{y}$. The following objective function is used to train the model:

$$L = \mathbb{E}_{\boldsymbol{y}, \boldsymbol{u} \sim \mathcal{N}(\mathbf{0}, \boldsymbol{I})} \left\| \boldsymbol{u} + \sigma_a \boldsymbol{s}\left(\boldsymbol{y} + \sigma_a \boldsymbol{u}; \theta\right) \right\|_2^2, \tag{18}$$

where $\sigma_a$ is a fixed value. Given $\sigma_a$, the optimal model $\boldsymbol{s}(\boldsymbol{y}; \theta^*)$ that minimizes $L$ is the score function of perturbed $\boldsymbol{y}$, $\boldsymbol{y} + \sigma_a \boldsymbol{u}$. In other words, we can approximate the score function $\boldsymbol{s}\left(\boldsymbol{y}\right)$ by using a sufficiently small $\sigma_a$. Related analysis can also be seen in [8, 23, 1]. During the training process, the value of $\sigma_a$ will be decreasing gradually to a very small value. This progressive process is helpful to numerically stabilize the model training. The only model training is to estimate $\boldsymbol{s}\left(\boldsymbol{y}\right)$ by $\boldsymbol{s}\left(\cdot; \theta\right)$, which is served as the first step of our approach. After the score function model is trained, we apply the denoising algorithms given in Section 3.1 to obtain denoised results and no more training is required.

## 4 Experiment

**Dataset and Implementation Details** We evaluate the proposed method for color images in the three benchmark datasets containing RGB natural images: Kodak dataset, CBSD68 [15] and CSet9. DIV2K [22] and CBSD500 dataset [3] are used as training datasets. The synthetic noise images for each noise model are generated and fixed through the training process. For the sake of fair comparison, we use the same modified U-Net [5] for all methods. When training, we randomly clip the training images to patches with the resolution of $128 \times 128$. AdamW optimizer [14] is used to train the network. We train each model for 5000 steps with the batch size of 32. To reduce memory, we utilize the tricks of cumulative gradient and mixed precision training. The learning rate is initialized to $1 \times 10^{-4}$ for first 4000 steps and it is decreased to $1 \times 10^{-5}$ for final 1000 steps. All the models are implemented in PyTorch [18] with NVidia V100. The pixel value range of all clean images is $[0, 255]$ and the parameters of noise models are built on it. Noisy images will be scaled when fed into the network. When an iterative algorithm is needed to solve Eq. 7, we set the number of iterations as 10. The more details of implementation are described in Supplementary Material.

**Baseline and Comparison Methods** We use supervised learning with MSE loss as the baseline model. Noisier2Noise and Neighbor2Neighbor are used as comparison methods. Since our approach

Table 1: The application range of different methods including supervised learning (SL), Noisier2Noise (Nr2N), Neighbor2Neighbor (Nb2Nb), Noise2Score (N2S) and ours. ✓ means applicable and × means not applicable or incapable to perform. For Neighbor2Neighbor, ✓$^*$ means that direct application is not feasible but indirect application is; ×$^*$ means that the application is not feasible but model training is executable.

| No. | Noise Model | SL | Nr2N | Nb2Nb | N2S | Ours |
|---|---|---|---|---|---|---|
| 1 | $\boldsymbol{y} = \boldsymbol{x} + \boldsymbol{\epsilon}, \boldsymbol{\epsilon} \sim \mathcal{N}\left(\mathbf{0}, \sigma^2 \boldsymbol{I}\right)$ | ✓ | ✓ | ✓ | ✓ | ✓ |
| 2 | $\boldsymbol{y} = \boldsymbol{x} + \boldsymbol{\epsilon}, \boldsymbol{\epsilon} \sim \mathcal{N}\left(\mathbf{0}, \sigma^2 \boldsymbol{A}^\top \boldsymbol{A}\right)$ | ✓ | ✓ | ×$^*$ | ✓ | ✓ |
| 3 | $\boldsymbol{y} = \boldsymbol{x} + \boldsymbol{\epsilon}, \boldsymbol{\epsilon} \sim \mathcal{N}\left(\mathbf{0}, \mathrm{diag}\left(a\boldsymbol{x} + b\mathbf{1}\right)^2\right)$ | ✓ | ✓ | ✓ | × | ✓ |
| 4 | $\boldsymbol{y} = \boldsymbol{x} + \boldsymbol{\epsilon}, \boldsymbol{\epsilon} \sim \mathcal{N}\left(\mathbf{0}, \boldsymbol{A}^\top \mathrm{diag}\left(a\boldsymbol{x} + b\mathbf{1}\right)^2 \boldsymbol{A}\right)$ | ✓ | ✓ | ×$^*$ | × | ✓ |
| 5 | $\boldsymbol{y} = \boldsymbol{\eta} \odot \boldsymbol{x}, \eta_i \sim \mathcal{G}\left(\alpha, \alpha\right)$ | ✓ | × | ✓ | ✓ | ✓ |
| 6 | $\boldsymbol{y} = \boldsymbol{A}\boldsymbol{\eta} \odot \boldsymbol{x}, \eta_i \sim \mathcal{G}\left(\alpha, \alpha\right)$ | ✓ | × | ✓$^*$ | ✓ | ✓ |
| 7 | $\boldsymbol{y} = \frac{1}{\lambda}\boldsymbol{\eta}, \eta_i \sim \mathcal{P}\left(\lambda x_i\right)$ | ✓ | × | ✓ | ✓ | ✓ |
| 8 | $\boldsymbol{y} = \frac{1}{\lambda}\boldsymbol{A}\boldsymbol{\eta}, \eta_i \sim \mathcal{P}\left(\lambda x_i\right)$ | ✓ | × | ✓$^*$ | ✓ | ✓ |
| 9 | $\boldsymbol{y} = (\boldsymbol{\eta} + \mathbf{1}) \odot \boldsymbol{x}, \eta_i \sim \mathcal{R}\left(\sigma\right)$ | ✓ | × | ×$^*$ | × | ✓ |
| 10 | $\boldsymbol{y} = \boldsymbol{A}(\boldsymbol{\eta} + \mathbf{1}) \odot \boldsymbol{x}, \eta_i \sim \mathcal{R}\left(\sigma\right)$ | ✓ | × | ×$^*$ | × | ✓ |
| 11 | $\boldsymbol{y} = \boldsymbol{\eta} \odot \boldsymbol{x} + \boldsymbol{\epsilon}, \eta_i \sim \mathcal{G}\left(\alpha, \alpha\right), \epsilon \sim \mathcal{N}\left(0, \sigma^2 \boldsymbol{I}\right)$ | ✓ | × | ✓ | × | ✓ |
| 12 | $\boldsymbol{y} = \boldsymbol{A}\boldsymbol{\eta} \odot \boldsymbol{x} + \boldsymbol{\epsilon}, \eta_i \sim \mathcal{G}\left(\alpha, \alpha\right), \epsilon \sim \mathcal{N}\left(0, \sigma^2 \boldsymbol{I}\right)$ | ✓ | × | ×$^*$ | × | ✓ |
| 13 | $\boldsymbol{y} = \frac{1}{\lambda}\boldsymbol{\eta} + \boldsymbol{\epsilon}, \eta_i \sim \mathcal{P}\left(\lambda x_i\right), \epsilon \sim \mathcal{N}\left(0, \sigma^2 \boldsymbol{I}\right)$ | ✓ | × | ✓ | × | ✓ |
| 14 | $\boldsymbol{y} = \frac{1}{\lambda}\boldsymbol{A}\boldsymbol{\eta} + \boldsymbol{\epsilon}, \eta_i \sim \mathcal{P}\left(\lambda x_i\right), \epsilon \sim \mathcal{N}\left(0, \sigma^2 \boldsymbol{I}\right)$ | ✓ | × | ×$^*$ | × | ✓ |
| 15 | $\boldsymbol{y} = (\boldsymbol{\eta} + \mathbf{1}) \odot \boldsymbol{x} + \boldsymbol{\epsilon}, \eta_i \sim \mathcal{R}\left(\sigma\right), \epsilon \sim \mathcal{N}\left(0, \sigma^2 \boldsymbol{I}\right)$ | ✓ | × | ×$^*$ | × | ✓ |
| 16 | $\boldsymbol{y} = \boldsymbol{A}(\boldsymbol{\eta} + \mathbf{1}) \odot \boldsymbol{x} + \boldsymbol{\epsilon}, \eta_i \sim \mathcal{R}\left(\sigma\right), \epsilon \sim \mathcal{N}\left(0, \sigma^2 \boldsymbol{I}\right)$ | ✓ | × | ×$^*$ | × | ✓ |

is identical to Noise2Score when the noise model follows exponential family distributions, we do not compare to it through metrics. Because Noisier2Noise can only be applied to additive noise, we do not train models by Noisier2Noise for other noise models. Though Neighbor2Neighbor is not suitable for some noise models from the perspective of theoretical analysis, we still train corresponding models and report its results. Table 1 shows the comparison of application range for different methods, including additive Gaussian noise, multiplicative noise and mixture noise. Based on it our experiments are conducted. Only supervised learning and our approach can handle all noise models listed in Table 1.

**Parameters of Noise Models**  Here, we emphasize that for all noise models in our experiments, $\boldsymbol{A}$ is set as a $3 \times 3$ convolution transform with the kernel of

$$\begin{pmatrix} 0.05 & 0.1 & 0.05 \\ 0.1 & 0.4 & 0.1 \\ 0.05 & 0.1 & 0.05 \end{pmatrix} \tag{19}$$

if it is used. The additive Gaussian noise in every mixture noise model is set as $\mathcal{N}\left(\mathbf{0}, 100\boldsymbol{I}\right)$. Other parameters will be given later. Finally, all parameters are assumed to be known in our experiments.

**Additive Gaussian Noise**  Using additive Gaussian noise, we consider four kinds of noise models with different $\boldsymbol{\Sigma}$ corresponding from No.1 to No.4 in Table 1. Our method is compared to supervised learning, Noisier2Noise and Neighbor2Neighbor as shown in Table 2. For the first two noise models $\sigma$ is 25, and for the rest $a = 0.98$ and $b = 25$. As expected, supervised learning performs best. In the cases without $\boldsymbol{A}$ Neighbor2Neighbor is the best among three other unsupervised learning methods. However, in the cases with $\boldsymbol{A}$ Neighbor2Neighbor performs very poorly. Our approach outperform other unsupervised learning methods in the second noise model and is competitive in the whole.

**Multiplicative Noise**  We consider the combination of three various multiplicative noise model (Gamma, Poisson and Rayleigh) and a convolution transform $\boldsymbol{A}$. They are corresponding from No.5 to No.10 in Table 1. Our method is compared to supervised learning and Neighbor2Neighbor and the results are shown in Table 3. Because Noisier2Noise can not address such multiplicative noise models, we neglect it. Though the noise is not pixel-wise independent when the convolution transform exists,

Table 2: Quantitative comparison for various parameters of $\Sigma$ in additive Gaussian noise using different methods in terms of PNSR (dB)/SSIM. Bold indicates the best result among three unsupervised methods, while underlined indicates the second-best result.

| | No.1: Gaussian, $\sigma = 25$ w/o $\boldsymbol{A}$ | | | No.2: Gaussian, $\sigma = 25$ w/ $\boldsymbol{A}$ | | |
|---|---|---|---|---|---|---|
| Method | Kodak | CSet9 | CBSD68 | Kodak | CSet9 | CBSD68 |
| SL | *32.44/0.887* | *30.27/0.891* | *31.32/0.892* | *34.80/0.928* | *32.53/0.924* | *34.08/0.938* |
| Nr2N | 31.80/0.863 | 29.75/0.878 | 30.84/0.871 | 33.87/**0.914** | 32.02/**0.916** | 33.40/**0.926** |
| Nb2Nb | **31.96/0.875** | **29.90/0.882** | **30.92/0.880** | 27.89/0.673 | 27.88/0.742 | 27.86/0.722 |
| Ours | 31.92/0.870 | 29.86/0.875 | 30.91/0.877 | **33.99/0.914** | **32.16**/0.914 | **33.45/0.926** |

| | No.3: Gaussian, $a = 0.98, b = 25$ w/o $\boldsymbol{A}$ | | | No.4: Gaussian, $a = 0.98, b = 25$ w/ $\boldsymbol{A}$ | | |
|---|---|---|---|---|---|---|
| Method | Kodak | CSet9 | CBSD68 | Kodak | CSet9 | CBSD68 |
| SL | *30.92/0.856* | *28.69/0.860* | *29.68/0.856* | *33.02/0.904* | *30.76/0.900* | *32.11/0.912* |
| Nr2N | 30.07/0.813 | 28.02/0.834 | 28.99/0.818 | **32.28/0.884** | **29.98/0.886** | **31.55/0.896** |
| Nb2Nb | **30.42/0.840** | **28.22/0.848** | **29.29/0.842** | 25.02/0.557 | 24.44/0.621 | 25.01/0.619 |
| Ours | 29.68/0.797 | 27.58/0.806 | 28.70/0.806 | 31.88/**0.884** | 29.50/0.874 | 31.14/0.895 |

Table 3: Quantitative comparison for various multiplicative noise models using different methods in terms of PNSR (dB)/SSIM. For Neighbor2Neighbor (Nb2Nb), if the noise model is constructed with $\boldsymbol{A}$, it can be regarded as the one without $\boldsymbol{A}$ through $\boldsymbol{A}^{-1}\boldsymbol{y}$. Thus we do not provide the metrics. Bold indicates the better result between two unsupervised methods.

| | No.5: Gamma, $\alpha = 26$ w/o $\boldsymbol{A}$ | | | No.6: Gamma, $\alpha = 26$ w/ $\boldsymbol{A}$ | | |
|---|---|---|---|---|---|---|
| Method | Kodak | CSet9 | CBSD68 | Kodak | CSet9 | CBSD68 |
| SL | *33.51/0.916* | *30.72/0.898* | *32.44/0.922* | *33.02/0.916* | *30.41/0.897* | *32.15/0.921* |
| Nb2Nb | **32.98/0.908** | **30.33/0.890** | **31.97/0.913** | - | - | - |
| Ours | 32.61/0.894 | 29.89/0.870 | 31.51/0.898 | 31.90/0.877 | 29.26/0.858 | 30.63/0.878 |

| | No.7: Poisson, $\lambda = 0.2$ w/o $\boldsymbol{A}$ | | | No.8: Poisson, $\lambda = 0.2$ w/ $\boldsymbol{A}$ | | |
|---|---|---|---|---|---|---|
| Method | Kodak | CSet9 | CBSD68 | Kodak | CSet9 | CBSD68 |
| SL | *32.90/0.902* | *30.56/0.894* | *31.87/0.908* | *32.55/0.902* | *30.27/0.894* | *31.64/0.907* |
| Nb2Nb | **32.50/0.893** | **30.17/0.886** | **31.48/0.899** | - | - | - |
| Ours | 32.38/0.886 | 29.98/0.874 | 31.31/0.891 | 31.84/0.872 | 29.48/0.864 | 30.56/0.873 |

| | No.9: Rayleigh, $\sigma = 0.3$ w/o $\boldsymbol{A}$ | | | No.10: Rayleigh, $\sigma = 0.3$ w/ $\boldsymbol{A}$ | | |
|---|---|---|---|---|---|---|
| Method | Kodak | CSet9 | CBSD68 | Kodak | CSet9 | CBSD68 |
| SL | *35.29/0.939* | *32.44/0.922* | *34.39/0.947* | *34.63/0.939* | *31.97/0.920* | *33.94/0.946* |
| Nb2Nb | 16.55/0.865 | 15.16/0.844 | 16.74/0.862 | - | - | - |
| Ours | **34.25/0.915** | **31.45/0.892** | **33.34/0.923** | **32.87/0.894** | **30.30/0.869** | **31.85/0.901** |

we can execute its inverse transform on $\boldsymbol{y}$ so that the requirement of pixel-wise independence is satisfied for Neighbor2Neighbor. Therefore, tackling noise models with a convolution transform is equivalent to the situations without $\boldsymbol{A}$. That is why we do not provide corresponding metrics result for Neighbor2Neighbor in Table 3. We set $\alpha$ as 26 for the Gamma noise, $\lambda$ as 0.2 for the Poisson noise, and $\sigma$ as 0.3 for the Rayleigh noise. When the noise model is based on Gamma or Poisson noise, it is unbiased, *i.e.* $\mathbb{E}[\boldsymbol{y} \mid \boldsymbol{x}] = \boldsymbol{x}$. In these cases, Neighbor2Neighbor is better than ours. However, when the noise model is based on Rayleigh noise which is biased our approach still has excellent performance while Neighbor2Neighbor is poor.

**Mixture Noise** We also consider the combination of three various multiplicative noise model (Gamma, Poisson and Rayleigh) and a convolution transform $\boldsymbol{A}$. For each one, additive Gaussian noise with $\Sigma = 100\boldsymbol{I}$ is added to construct mixture noise models. They are corresponding from No.11 to No.16 in Table 1. Because of the same reason discussed before, our method is compared to supervised learning and Neighbor2Neighbor, and Noisier2Noise is neglected. The experimental

Table 4: Quantitative comparison for various mixture noise models using different methods in terms of PNSR (dB)/SSIM. For each noise model, additive Gaussian noise with $\Sigma = 100\boldsymbol{I}$ is added. Bold indicates the better result between two unsupervised methods.

| | No.11: Gamma, $\alpha = 26$ w/o $\boldsymbol{A}$ | | | No.12: Gamma, $\alpha = 26$ w/ $\boldsymbol{A}$ | | |
|---|---|---|---|---|---|---|
| Method | Kodak | CSet9 | CBSD68 | Kodak | CSet9 | CBSD68 |
| SL | *32.86/0.902* | *30.23/0.889* | *31.70/0.906* | *33.02/0.882* | *29.30/0.875* | *30.42/0.882* |
| Nb2Nb | **32.30/0.892** | **29.80/0.880** | **31.21/0.896** | 26.93/0.676 | 25.06/0.663 | 26.33/0.710 |
| Ours | 32.13/0.880 | 29.61/0.861 | 31.08/0.887 | **30.74/0.853** | **28.40/0.841** | **29.58/0.854** |
| | No.13: Poisson, $\lambda = 0.2$ w/o $\boldsymbol{A}$ | | | No.14: Poisson, $\lambda = 0.2$ w/ $\boldsymbol{A}$ | | |
| Method | Kodak | CSet9 | CBSD68 | Kodak | CSet9 | CBSD68 |
| SL | *32.49/0.892* | *30.18/0.887* | *31.40/0.897* | *31.44/0.877* | *29.36/0.876* | *30.32/0.878* |
| Nb2Nb | 32.03/**0.884** | **29.74/0.880** | 30.98/**0.888** | 26.82/0.646 | 25.48/0.669 | 26.32/0.688 |
| Ours | **32.10**/0.879 | 29.70/0.870 | **31.03**/0.883 | **31.11/0.855** | **28.80/0.851** | **29.85/0.856** |
| | No.15: Rayleigh, $\sigma = 0.3$ w/o $\boldsymbol{A}$ | | | No.16: Rayleigh, $\sigma = 0.3$ w/ $\boldsymbol{A}$ | | |
| Method | Kodak | CSet9 | CBSD68 | Kodak | CSet9 | CBSD68 |
| SL | *34.38/0.926* | *31.79/0.913* | *33.39/0.933* | *32.83/0.908* | *30.48/0.898* | *31.74/0.910* |
| Nb2Nb | 16.54/0.856 | 15.15/0.838 | 16.71/0.851 | 16.40/0.704 | 15.06/0.738 | 16.56/0.736 |
| Ours | **33.54/0.902** | **30.94/0.883** | **32.68/0.913** | **31.14/0.867** | **28.94/0.847** | **30.23/0.876** |

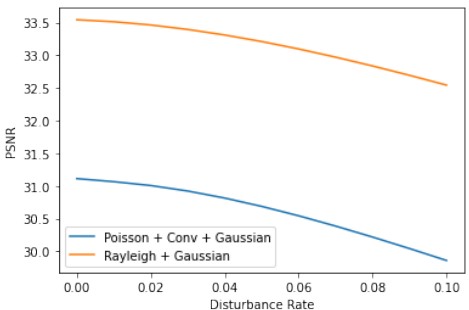

Figure 2: PSNR V.S. disturbance rate for No.14 and No.15 noise models.

results are shown in Table 4. Due to the additive Gaussian Noise, Neighbor2Neighbor are not able to handle the cases with $\boldsymbol{A}$ through the inverse convolution transform. The correlation of noise hampers the performance of Neigh2bor2Neighbor. Except the first noise model in Table 4, our approach all outperforms Neighbor2Neighbor and achieve excellent performance near to supervised learning.

**Robustness Evaluation** To apply our approach, the parameters of noise models have to be known beforehand. Therefore their precision may impact on the performance. We choose No.14 and No.15 noise models in Table 1 as examples to show the robustness to parameters' precision. Suppose $k$ is one of parameters, we disturb it by $(a + 1)k$ where $a \sim \mathcal{N}(0, r^2)$ and $r$ is called the disturbance rate. The larger $r$, the less precise the noise model. As $r$ is increasing, the PSNR of the denoised result is shown in Figure 2. When $r = 0.1$, the PSNR reduces about 1 dB, which displays the robustness.

## 5   Conclusion

In this paper, we propose a new approach for unsupervised image denoising. The key part is Proposition 3.1. Based on it, we construct an equation, Eq. 7 about the clean image $\boldsymbol{x}$ and the noisy image $\boldsymbol{y}$. After the score function of $\boldsymbol{y}$ is estimated, the denoised result can be obtained by solving the equation. Our approach can be applied to many different noise model as long as Eq. 7 is solvable. The denoising performance is competitive for simple noise models and excellent for complicated ones. We hope that this work is helpful to address sophisticated image denoising problems in practice.

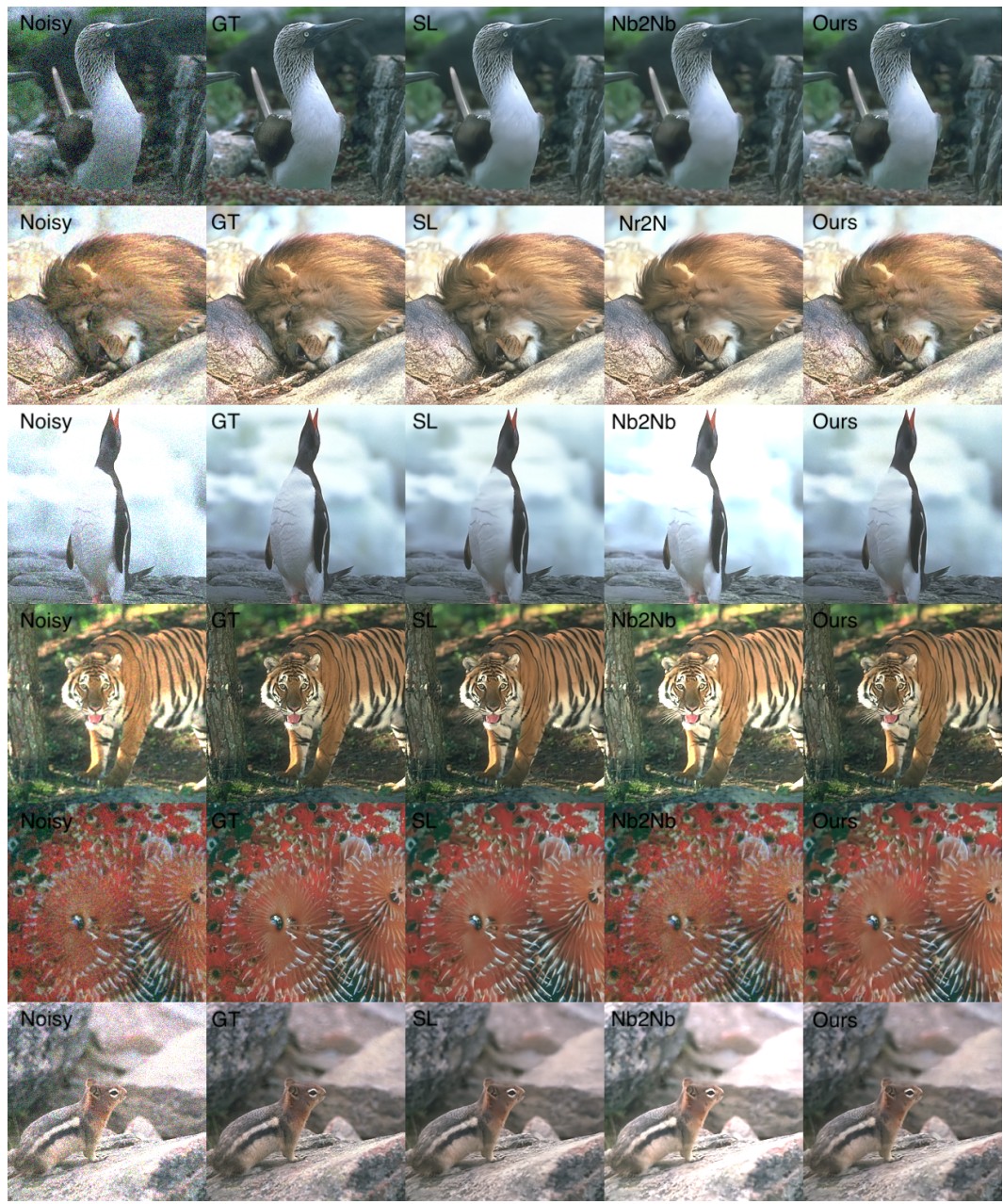

Figure 3: Qualitative Comparison using CBSD68 dataset (cropped to $256 \times 256$). From the first row to the last: (1) Gaussian noise, $\sigma = 25$ w/o $\boldsymbol{A}$; (2) Gaussian noise, $\sigma = 25$ w/ $\boldsymbol{A}$; (3) Rayleigh noise, $\sigma = 0.3$ w/o $\boldsymbol{A}$; (4) Rayleigh noise, $\sigma = 0.3$ w/ $\boldsymbol{A}$; (5) Poisson noise, $\lambda = 0.2$ w/ $\boldsymbol{A}$ added by Gaussian Noise with $\sigma = 10$; (6) Rayleigh noise, $\sigma = 0.3$ w/o $\boldsymbol{A}$ added by Gaussian Noise with $\sigma = 10$. Noisy: noisy image, GT: ground-truth image, SL: supervised learning, Nb2Nb: Neighbor2Neighbor, Nr2N: Noisier2Noise.

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
