# Supplementary Material

## 1   Proofs and Derivations

### 1.1   The Proof of Eq. 4 in Section 3.1.1

*Proof.*

$$\nabla_{\boldsymbol{y}} \log p\left(\boldsymbol{y}\right) = \int p\left(\boldsymbol{x} \mid \boldsymbol{y}\right) \nabla_{\boldsymbol{y}} \log p\left(\boldsymbol{y} \mid \boldsymbol{x}\right) \mathrm{d}\boldsymbol{x}$$

$$= \int p\left(\boldsymbol{x} \mid \boldsymbol{y}\right) \left(\nabla_{\boldsymbol{y}} \log b\left(\boldsymbol{y}\right) + \nabla_{\boldsymbol{y}} H(\boldsymbol{x})^{\top} T\left(\boldsymbol{y}\right)\right) \mathrm{d}\boldsymbol{x}$$

$$= \nabla_{\boldsymbol{y}} \log b\left(\boldsymbol{y}\right) + T'\left(\boldsymbol{y}\right)^{\top} \int p\left(\boldsymbol{x} \mid \boldsymbol{y}\right) H(\boldsymbol{x}) \mathrm{d}\boldsymbol{x}$$

$$= \nabla_{\boldsymbol{y}} \log b\left(\boldsymbol{y}\right) + T'\left(\boldsymbol{y}\right)^{\top} \mathbb{E}\left[H(\boldsymbol{x}) \mid \boldsymbol{y}\right].$$

$\square$

### 1.2   The Proof of Proposition 3.1 in Section 3.1.2

*Proof.*  Our derivation begins with the right part of Eq. 6:

$$\int p\left(\boldsymbol{x} \mid \boldsymbol{y}\right) \nabla_{\boldsymbol{y}} \log p\left(\boldsymbol{y} \mid \boldsymbol{x}\right) \mathrm{d}\boldsymbol{x}$$

$$= \int p\left(\boldsymbol{x} \mid \boldsymbol{y}\right) \nabla_{\boldsymbol{y}} \log \frac{p\left(\boldsymbol{x} \mid \boldsymbol{y}\right) p\left(\boldsymbol{y}\right)}{p\left(\boldsymbol{x}\right)} \mathrm{d}\boldsymbol{x}$$

$$= \int p\left(\boldsymbol{x} \mid \boldsymbol{y}\right) \left[\nabla_{\boldsymbol{y}} \log p\left(\boldsymbol{x} \mid \boldsymbol{y}\right) + \nabla_{\boldsymbol{y}} \log p\left(\boldsymbol{y}\right) - \nabla_{\boldsymbol{y}} \log p\left(\boldsymbol{x}\right)\right] \mathrm{d}\boldsymbol{x}$$

$$= \int p\left(\boldsymbol{x} \mid \boldsymbol{y}\right) \left[\nabla_{\boldsymbol{y}} \log p\left(\boldsymbol{x} \mid \boldsymbol{y}\right) + \nabla_{\boldsymbol{y}} \log p\left(\boldsymbol{y}\right)\right] \mathrm{d}\boldsymbol{x}$$

$$= \int p\left(\boldsymbol{x} \mid \boldsymbol{y}\right) \nabla_{\boldsymbol{y}} \log p\left(\boldsymbol{x} \mid \boldsymbol{y}\right) \mathrm{d}\boldsymbol{x} + \nabla_{\boldsymbol{y}} \log p\left(\boldsymbol{y}\right) \int p\left(\boldsymbol{x} \mid \boldsymbol{y}\right) \mathrm{d}\boldsymbol{x}$$

$$= \int p\left(\boldsymbol{x} \mid \boldsymbol{y}\right) \nabla_{\boldsymbol{y}} \log p\left(\boldsymbol{x} \mid \boldsymbol{y}\right) \mathrm{d}\boldsymbol{x} + \nabla_{\boldsymbol{y}} \log p\left(\boldsymbol{y}\right).$$

6   Now, we prove that $\int p\left(\boldsymbol{x}\mid\boldsymbol{y}\right)\nabla_{\boldsymbol{y}}\log p\left(\boldsymbol{x}\mid\boldsymbol{y}\right)\mathrm{d}\boldsymbol{x}=0$:

$$
\int p\left(\boldsymbol{x}\mid\boldsymbol{y}\right)\nabla_{\boldsymbol{y}}\log p\left(\boldsymbol{x}\mid\boldsymbol{y}\right)\mathrm{d}\boldsymbol{x}
$$

$$
=\int p\left(\boldsymbol{x}\mid\boldsymbol{y}\right)\frac{1}{p\left(\boldsymbol{x}\mid\boldsymbol{y}\right)}\nabla_{\boldsymbol{y}}p\left(\boldsymbol{x}\mid\boldsymbol{y}\right)\mathrm{d}\boldsymbol{x}
$$

$$
=\int \nabla_{\boldsymbol{y}}p\left(\boldsymbol{x}\mid\boldsymbol{y}\right)\mathrm{d}\boldsymbol{x}
$$

$$
=\nabla_{\boldsymbol{y}}\int p\left(\boldsymbol{x}\mid\boldsymbol{y}\right)\mathrm{d}\boldsymbol{x}
$$

$$
=\nabla_{\boldsymbol{y}}1=0.
$$

7   Thus, Eq. 6 is proved.                                                                                              $\square$

## 1.3   The Proof of Theorem 3.1 in Section 3.1.2

9   *Proof.* Given $\boldsymbol{y}$ suppose $\boldsymbol{f}(\boldsymbol{x},\boldsymbol{y})$ is invertible. Denote its inverse function as $\boldsymbol{f}_{\boldsymbol{y}}^{-1}$. By solving Eq. 7,
10  we obtain that

$$
\hat{\boldsymbol{x}}=\boldsymbol{f}_{\boldsymbol{y}}^{-1}\left(\boldsymbol{s}\left(\boldsymbol{y}\right)\right)=\boldsymbol{f}_{\boldsymbol{y}}^{-1}\left(\mathbb{E}_{\boldsymbol{x}\mid\boldsymbol{y}}\left[\boldsymbol{f}\left(\boldsymbol{x},\boldsymbol{y}\right)\right]\right).
$$

11  Let the Lipschitz constant of $\boldsymbol{f}_{\boldsymbol{y}}^{-1}$ is $L_{\boldsymbol{f}_{\boldsymbol{y}}^{-1}}$, the Hessian matrix of $\boldsymbol{f}_i$ is $\boldsymbol{H}_{\boldsymbol{f}_i}$ and $n$ is the dimension
12  of $\boldsymbol{x}$. Then, we can derive that:

$$
\left\|\boldsymbol{f}_{\boldsymbol{y}}^{-1}\left(\mathbb{E}_{\boldsymbol{x}\mid\boldsymbol{y}}\left[\boldsymbol{f}\left(\boldsymbol{x},\boldsymbol{y}\right)\right]\right)-\mathbb{E}_{\boldsymbol{x}\mid\boldsymbol{y}}\left[\boldsymbol{x}\right]\right\|_2
$$

$$
=\left\|\boldsymbol{f}_{\boldsymbol{y}}^{-1}\left(\mathbb{E}_{\boldsymbol{x}\mid\boldsymbol{y}}\left[\boldsymbol{f}\left(\boldsymbol{x},\boldsymbol{y}\right)\right]\right)-\boldsymbol{f}_{\boldsymbol{y}}^{-1}\left(\boldsymbol{f}\left(\mathbb{E}_{\boldsymbol{x}\mid\boldsymbol{y}}[\boldsymbol{x}],\boldsymbol{y}\right)\right)\right\|_2
$$

$$
\leq L_{\boldsymbol{f}_{\boldsymbol{y}}^{-1}}\left\|\mathbb{E}_{\boldsymbol{x}\mid\boldsymbol{y}}\left[\boldsymbol{f}(\boldsymbol{x},\boldsymbol{y})\right]-\boldsymbol{f}\left(\mathbb{E}_{\boldsymbol{x}\mid\boldsymbol{y}}\left[\boldsymbol{x}\right],\boldsymbol{y}\right)\right\|_2
$$

$$
=L_{\boldsymbol{f}_{\boldsymbol{y}}^{-1}}\left(\sum_{i=1}^{n}\left(\mathbb{E}_{\boldsymbol{x}\mid\boldsymbol{y}}\left[\boldsymbol{f}\left(\boldsymbol{x},\boldsymbol{y}\right)_i\right]-\boldsymbol{f}\left(\mathbb{E}_{\boldsymbol{x}\mid\boldsymbol{y}}\left[\boldsymbol{x}\right],\boldsymbol{y}\right)_i\right)^2\right)^{1/2}
$$

$$
=L_{\boldsymbol{f}_{\boldsymbol{y}}^{-1}}\left(\sum_{i=1}^{n}\left(\mathbb{E}_{\boldsymbol{x}\mid\boldsymbol{y}}\left[\boldsymbol{f}\left(\mathbb{E}_{\boldsymbol{x}\mid\boldsymbol{y}}\left[\boldsymbol{x}\right],\boldsymbol{y}\right)_i+\left(\boldsymbol{x}-\mathbb{E}_{\boldsymbol{x}\mid\boldsymbol{y}}\left[\boldsymbol{x}\right]\right)^{\top}\nabla\boldsymbol{f}\left(\mathbb{E}_{\boldsymbol{x}\mid\boldsymbol{y}}\left[\boldsymbol{x}\right]\right)_i\right.\right.
$$

$$
+\frac{1}{2}\left(\boldsymbol{x}-\mathbb{E}_{\boldsymbol{x}\mid\boldsymbol{y}}\left[\boldsymbol{x}\right]\right)^{\top}\boldsymbol{H}_{\boldsymbol{f}_i}\left(\mathbb{E}_{\boldsymbol{x}\mid\boldsymbol{y}}\left[\boldsymbol{x}\right]\right)\left(\boldsymbol{x}-\mathbb{E}_{\boldsymbol{x}\mid\boldsymbol{y}}\left[\boldsymbol{x}\right]\right)
$$

$$
\left.\left.+o\left(\left\|\boldsymbol{x}-\mathbb{E}_{\boldsymbol{x}\mid\boldsymbol{y}}\left[\boldsymbol{x}\right]\right\|_2^2\right)\right]-\boldsymbol{f}\left(\mathbb{E}_{\boldsymbol{x}\mid\boldsymbol{y}}\left[\boldsymbol{x}\right]\right)_i\right)^2\right)^{1/2}
$$

$$
=L_{\boldsymbol{f}_{\boldsymbol{y}}^{-1}}\left(\sum_{i=1}^{n}\left(\mathbb{E}_{\boldsymbol{x}\mid\boldsymbol{y}}\left[\frac{1}{2}\left(\boldsymbol{x}-\mathbb{E}_{\boldsymbol{x}\mid\boldsymbol{y}}\left[\boldsymbol{x}\right]\right)^{\top}\boldsymbol{H}_{\boldsymbol{f}_i}\left(\mathbb{E}_{\boldsymbol{x}\mid\boldsymbol{y}}\left[\boldsymbol{x}\right]\right)\left(\boldsymbol{x}-\mathbb{E}_{\boldsymbol{x}\mid\boldsymbol{y}}\left[\boldsymbol{x}\right]\right)+o\left(\left\|\boldsymbol{x}-\mathbb{E}_{\boldsymbol{x}\mid\boldsymbol{y}}\left[\boldsymbol{x}\right]\right\|_2^2\right)\right]\right)^2\right)^{1/2}
$$

$$
\leq L_{\boldsymbol{f}_{\boldsymbol{y}}^{-1}}\left(\sum_{i=1}^{n}\left(\mathbb{E}_{\boldsymbol{x}\mid\boldsymbol{y}}\left[\frac{1}{2}H_{max}\left\|\boldsymbol{x}-\mathbb{E}_{\boldsymbol{x}\mid\boldsymbol{y}}\left[\boldsymbol{x}\right]\right\|_2^2\right]+o\left(\mathbb{E}_{\boldsymbol{x}\mid\boldsymbol{y}}\left[\left\|\boldsymbol{x}-\mathbb{E}_{\boldsymbol{x}\mid\boldsymbol{y}}\left[\boldsymbol{x}\right]\right\|_2^2\right]\right)\right)^2\right)^{1/2}
$$

$$
=\sqrt{n}L_{\boldsymbol{f}_{\boldsymbol{y}}^{-1}}\left(\frac{1}{2}H_{max}\mathbb{E}_{\boldsymbol{x}\mid\boldsymbol{y}}\left[\left\|\boldsymbol{x}-\mathbb{E}_{\boldsymbol{x}\mid\boldsymbol{y}}\left[\boldsymbol{x}\right]\right\|_2^2\right]+o\left(\mathbb{E}_{\boldsymbol{x}\mid\boldsymbol{y}}\left[\left\|\boldsymbol{x}-\mathbb{E}_{\boldsymbol{x}\mid\boldsymbol{y}}\left[\boldsymbol{x}\right]\right\|_2^2\right]\right)\right)
$$

$$
=\sqrt{n}L_{\boldsymbol{f}_{\boldsymbol{y}}^{-1}}\left(H_{max}\mathrm{Tr}\left(\mathrm{Cov}\left[\boldsymbol{x}\mid\boldsymbol{y}\right]\right)+o\left(\mathrm{Tr}\left(\mathrm{Cov}\left[\boldsymbol{x}\mid\boldsymbol{y}\right]\right)\right)\right),
$$

13  where $H_{max}$ is the maximal value of $\boldsymbol{H}_{\boldsymbol{f}_i}$ for any $i$. Let $\frac{1}{2}\sqrt{n}L_{\boldsymbol{f}_{\boldsymbol{y}}^{-1}}H_{max}=C$, we prove Eq. 8 in
14  Theorem 3.1.

15                                                                                                                     $\square$

 **1.4    A Useful Lemma**

 **Lemma 1.1.** *We state the probability density transform equation as follows. Suppose $\boldsymbol{x} \sim \mathbf{x}$ and*
 *$\boldsymbol{y} \sim \mathbf{y}$ and $\boldsymbol{y} = f(\boldsymbol{x})$. Assume $f$ is invertible and its inverse function is $g$. Then, we have*

$$p_{\mathbf{y}}\left(\boldsymbol{y}\right) = \left|\frac{\partial g}{\partial \boldsymbol{y}}\right| p_{\mathbf{x}}\left(g(\boldsymbol{y})\right).$$

 **1.5    The Derivation of $\boldsymbol{f}\left(\boldsymbol{x}, \boldsymbol{y}\right)$ for Gamma Noise in Section 3.2.2**

 The derivation target is:

$$\boldsymbol{f}\left(\boldsymbol{x}, \boldsymbol{y}\right) = \nabla_{\boldsymbol{y}} \log p\left(\boldsymbol{y} \mid \boldsymbol{x}\right) = \frac{\alpha - 1}{\boldsymbol{y}} - \frac{\alpha}{\boldsymbol{x}}.$$

 *Proof.*  According to Eq. 11, we have that:

$$
\begin{aligned}
\nabla_{\boldsymbol{y}} \log p\left(\boldsymbol{y} \mid \boldsymbol{x}\right) &= \nabla_{\boldsymbol{y}} \log \prod_{i=1}^{d} \frac{\alpha^{\alpha}}{\Gamma\left(\alpha\right)} \left(\frac{y_i}{x_i}\right)^{\alpha - 1} \exp\left\{-\frac{\alpha y_i}{x_i}\right\} \cdot \frac{1}{x_i} \\
&= \nabla_{\boldsymbol{y}} \sum_{i=1}^{d} \log \frac{\alpha^{\alpha}}{\Gamma\left(\alpha\right)} \left(\frac{y_i}{x_i}\right)^{\alpha - 1} \exp\left\{-\frac{\alpha y_i}{x_i}\right\} \cdot \frac{1}{x_i} \\
&= \sum_{i=1}^{d} \nabla_{\boldsymbol{y}} \log \frac{\alpha^{\alpha}}{\Gamma\left(\alpha\right)} \left(\frac{y_i}{x_i}\right)^{\alpha - 1} \exp\left\{-\frac{\alpha y_i}{x_i}\right\} \cdot \frac{1}{x_i} \\
&= \sum_{i=1}^{d} \nabla_{\boldsymbol{y}} \left((\alpha - 1) \log y_i - \frac{\alpha y_i}{x_i}\right) \\
&= \frac{\alpha - 1}{\boldsymbol{y}} - \frac{\alpha}{\boldsymbol{x}}.
\end{aligned}
$$

 $\square$

 **1.6    The Proof of Eq. 12 in Section 3.2.2**

*Proof.*

$$
\begin{aligned}
\boldsymbol{s}\left(\boldsymbol{y}\right) &= \frac{\alpha - 1}{\boldsymbol{y}} - \frac{\alpha}{\boldsymbol{x}} \\
\Longleftrightarrow \boldsymbol{y} \odot \boldsymbol{s}\left(\boldsymbol{y}\right) &= \alpha - 1 - \frac{\alpha \boldsymbol{y}}{\boldsymbol{x}} \\
\Longleftrightarrow \frac{\alpha \boldsymbol{y}}{\boldsymbol{x}} &= \alpha - 1 - \boldsymbol{y} \odot \boldsymbol{s}\left(\boldsymbol{y}\right) \\
\Longleftrightarrow \boldsymbol{x} &= \frac{\alpha \boldsymbol{y}}{\alpha - 1 - \boldsymbol{y} \odot \boldsymbol{s}\left(\boldsymbol{y}\right)}.
\end{aligned}
$$

 $\square$

 **1.7    The Derivation of $\boldsymbol{f}\left(\boldsymbol{x}, \boldsymbol{y}\right)$ for Poisson Noise in Section 3.2.2**

 The derivation target is:

$$\boldsymbol{f}\left(\boldsymbol{x}, \boldsymbol{y}\right) = \nabla_{\boldsymbol{y}} \log \Pr\left(\boldsymbol{y} \mid \boldsymbol{x}\right) = \lambda \log\left(\lambda \boldsymbol{x}\right) - \lambda \log\left(\lambda \boldsymbol{y} + \frac{1}{2}\right).$$

27    *Proof.* According to Eq. 13, we have that

$$
\begin{aligned}
\nabla_{\boldsymbol{y}} \log \Pr\left(\boldsymbol{y} \mid \boldsymbol{x}\right) =& \nabla_{\boldsymbol{y}} \log \prod_{i=1}^{d} \frac{\left(\lambda x_i\right)^{\lambda_i y_i}}{\left(\lambda y_i\right)!} e^{-\lambda x_i} \\
=& \sum_{i=1}^{d} \nabla_{\boldsymbol{y}} \log \frac{\left(\lambda x_i\right)^{\lambda_i y_i}}{\left(\lambda y_i\right)!} e^{-\lambda x_i} \\
=& \sum_{i=1}^{d} \nabla_{\boldsymbol{y}} \left(\lambda_i y_i \log \lambda x_i - \log\left(\lambda y_i\right)!\right) \\
=& \lambda \log\left(\lambda \boldsymbol{x}\right) - \lambda \log\left(\lambda \boldsymbol{y} + \frac{1}{2}\right).
\end{aligned}
$$

28    Here, we set $\nabla_{y_i} \log\left(\lambda y_i\right)! = \lambda \log\left(\lambda y_i + \frac{1}{2}\right)$.     □

29    **1.8    The Proof of Eq. 14 in Section 3.2.2**

*Proof.*

$$
\begin{aligned}
\boldsymbol{s}\left(\boldsymbol{y}\right) =& \lambda \log\left(\lambda \boldsymbol{x}\right) - \lambda \log\left(\lambda \boldsymbol{y} + \frac{1}{2}\right) \\
\Longleftrightarrow \frac{\boldsymbol{s}\left(\boldsymbol{y}\right)}{\lambda} =& \log\left(\lambda \boldsymbol{x}\right) - \log\left(\lambda \boldsymbol{y} + \frac{1}{2}\right) \\
\Longleftrightarrow \log\left(\lambda \boldsymbol{x}\right) =& \frac{\boldsymbol{s}\left(\boldsymbol{y}\right)}{\lambda} + \log\left(\lambda \boldsymbol{y} + \frac{1}{2}\right) \\
\Longleftrightarrow \lambda \boldsymbol{x} =& \left(\lambda \boldsymbol{y} + \frac{1}{2}\right) \odot \exp\left\{\frac{\boldsymbol{s}\left(\boldsymbol{y}\right)}{\lambda}\right\} \\
\Longleftrightarrow \boldsymbol{x} =& \left(\boldsymbol{y} + \frac{1}{2\lambda}\right) \odot \exp\left\{\frac{\boldsymbol{s}\left(\boldsymbol{y}\right)}{\lambda}\right\}.
\end{aligned}
$$

30         □

31    **1.9    The Derivation of $\boldsymbol{f}\left(\boldsymbol{x}, \boldsymbol{y}\right)$ for Rayleigh Noise in Section 3.2.2**

32    The derivation target is:

$$
\boldsymbol{f}\left(\boldsymbol{x}, \boldsymbol{y}\right) = \nabla_{\boldsymbol{y}} \log p\left(\boldsymbol{y} \mid \boldsymbol{x}\right) = \frac{1}{\boldsymbol{y} - \boldsymbol{x}} - \frac{\boldsymbol{y} - \boldsymbol{x}}{\sigma^2 \boldsymbol{x}^2}.
$$

33    *Proof.* According to Eq. 15, we have that

$$
\begin{aligned}
\nabla_{\boldsymbol{y}} \log p\left(\boldsymbol{y} \mid \boldsymbol{x}\right) =& \nabla_{\boldsymbol{y}} \log \prod_{i=1}^{d} \frac{1}{x_i} \frac{y_i - x_i}{x_i \sigma^2} \exp\left\{-\frac{\left(y_i - x_i\right)^2}{2 x_i^2 \sigma^2}\right\} \\
=& \sum_{i=1}^{d} \nabla_{\boldsymbol{y}} \log \frac{1}{x_i} \frac{y_i - x_i}{x_i \sigma^2} \exp\left\{-\frac{\left(y_i - x_i\right)^2}{2 x_i^2 \sigma^2}\right\} \\
=& \sum_{i=1}^{d} \nabla_{\boldsymbol{y}} \left(\log\left(y_i - x_i\right) - \frac{\left(y_i - x_i\right)^2}{2 x_i^2 \sigma^2}\right) \\
=& \frac{1}{\boldsymbol{y} - \boldsymbol{x}} - \frac{\boldsymbol{y} - \boldsymbol{x}}{\sigma^2 \boldsymbol{x}^2}.
\end{aligned}
$$

34         □

## 1.10 The Proof of the Solving Method in Algorithm 3

*Proof.* Our target equation is:

$$\boldsymbol{s}\left(\boldsymbol{y}\right) = \frac{1}{\boldsymbol{y}-\boldsymbol{x}} - \frac{\boldsymbol{y}-\boldsymbol{x}}{\sigma^2 \boldsymbol{x}^2}.$$

For simplicity, we do not use bold font. Let $t = \frac{y-x}{x}$ and assume $t > 0$ because $x$ should be smaller than $y$ according to the Rayleigh distribution. We denote $\boldsymbol{s}\left(\boldsymbol{y}\right)$ as $s$. Fixing $x$, then

$$s = \frac{1}{y-x} - \frac{y-x}{\sigma^2 x^2}$$

$$\Longleftrightarrow sx = \frac{x}{y-x} - \frac{y-x}{\sigma^2 x}$$

$$\Longleftrightarrow sx = \frac{1}{t} - \frac{t}{\sigma^2}$$

$$\Longleftrightarrow t^2 + \sigma^2 sxt - \sigma^2 = 0$$

Since $t > 0$, we have

$$t = \frac{-\sigma^2 sx + \sqrt{\sigma^4 s^2 x^2 + 4\sigma^2}}{2}$$

After solving $t$, we compute $x = \frac{y}{t+1}$. Therefore, the iterative process contains two steps:

- $t = \frac{-\sigma^2 sx + \sqrt{\sigma^4 s^2 x^2 + 4\sigma^2}}{2}$.

- $x = \frac{y}{t+1}$.

$\square$

## 1.11 The Derivation of $\nabla_{\boldsymbol{y}} \log p_{\mathbf{y}}\left(\boldsymbol{y} \mid \boldsymbol{x}\right)$ for Multiplicative Noise with Convolution Transform in Section 3.2.2

The derivation target is:

$$\nabla_{\boldsymbol{y}} \log p_{\mathbf{y}}\left(\boldsymbol{y} \mid \boldsymbol{x}\right) = \boldsymbol{A}^{-1,\top} \nabla_{\boldsymbol{z}} \log p_{\mathbf{z}}\left(\boldsymbol{A}^{-1}\boldsymbol{y} \mid \boldsymbol{x}\right).$$

*Proof.* According to Lemma 1.1, we have $\boldsymbol{y} = f(\boldsymbol{z}) = \boldsymbol{A}^{-1}\boldsymbol{z}$, then $g(\boldsymbol{y}) = \boldsymbol{A}\boldsymbol{y}$. Thus,

$$p_{\mathbf{y}}\left(\boldsymbol{y} \mid \boldsymbol{x}\right) = \left|\boldsymbol{A}^{-1}\right| \nabla_{\boldsymbol{z}} \log p_{\mathbf{z}}\left(\boldsymbol{A}^{-1}\boldsymbol{y} \mid \boldsymbol{x}\right)$$

$$\Longleftrightarrow \nabla_{\boldsymbol{y}} \log p_{\mathbf{y}}\left(\boldsymbol{y} \mid \boldsymbol{x}\right) = \boldsymbol{A}^{-1,\top} \nabla_{\boldsymbol{z}} \log p_{\mathbf{z}}\left(\boldsymbol{A}^{-1}\boldsymbol{y} \mid \boldsymbol{x}\right).$$

$\square$

## 1.12 The Proof of Eq. 16 in Section 3.2.3

*Proof.* Let $\bar{\boldsymbol{z}} = \mathbb{E}\left[\boldsymbol{z} \mid \boldsymbol{y}\right]$. Then, we have:

$$p_{\mathbf{y}}\left(\boldsymbol{y} \mid \boldsymbol{x}\right) \approx \int_{\boldsymbol{z} \approx \boldsymbol{y}} p_{\mathbf{y}}\left(\boldsymbol{y} \mid \boldsymbol{z}\right) p_{\mathbf{z}}\left(\boldsymbol{z} \mid \boldsymbol{x}\right) \mathrm{d}\boldsymbol{z}$$

$$\approx \int_{\boldsymbol{z} \approx \boldsymbol{y}} p_{\mathbf{y}}\left(\boldsymbol{y} \mid \boldsymbol{z}\right) \left(p_{\mathbf{z}}\left(\bar{\boldsymbol{z}} \mid \boldsymbol{x}\right) + \nabla_{\boldsymbol{z}} p_{\mathbf{z}}\left(\bar{\boldsymbol{z}} \mid \boldsymbol{x}\right)^T \left(\boldsymbol{z} - \bar{\boldsymbol{z}}\right)\right) \mathrm{d}\boldsymbol{z}$$

$$= p_{\mathbf{z}}\left(\bar{\boldsymbol{z}} \mid \boldsymbol{x}\right) \int_{\boldsymbol{z} \approx \boldsymbol{y}} p_{\mathbf{y}}\left(\boldsymbol{y} \mid \boldsymbol{z}\right) \mathrm{d}\boldsymbol{z} + \nabla_{\boldsymbol{z}} p_{\mathbf{z}}\left(\bar{\boldsymbol{z}} \mid \boldsymbol{x}\right)^T \int_{\boldsymbol{z} \approx \boldsymbol{y}} p_{\mathbf{y}}\left(\boldsymbol{y} \mid \boldsymbol{z}\right)\left(\boldsymbol{z} - \bar{\boldsymbol{z}}\right) \mathrm{d}\boldsymbol{z}$$

$$\approx p_{\mathbf{z}}\left(\bar{\boldsymbol{z}} \mid \mathbf{x}\right) + \nabla_{\boldsymbol{z}} p_{\mathbf{z}}\left(\bar{\boldsymbol{z}} \mid \boldsymbol{x}\right)^T \left(\boldsymbol{y} - \bar{\boldsymbol{z}}\right).$$

$\square$

Table 1: The specific conclusions of Gaussian, Gamma and Poisson noise in Noise2Score.

| Noise | $H(\boldsymbol{x})$ | $T(\boldsymbol{y})$ | $b(\boldsymbol{y})$ | $H^{-1}_{T(\boldsymbol{y})}(\boldsymbol{z})$ | $\hat{x}$ |
|---|---|---|---|---|---|
| Gaussian | $\dfrac{\boldsymbol{x}}{\sigma^2}$ | $\boldsymbol{y}$ | $\dfrac{1}{\sqrt{2\pi}^d \sigma^d} e^{-\frac{\|\boldsymbol{y}\|_2^2}{2\sigma^2}}$ | $\sigma^2 \boldsymbol{z}$ | $\sigma^2 \boldsymbol{s}(\boldsymbol{y}) + \boldsymbol{y}$ |
| Gamma | $\left(\alpha\mathbf{1}-\mathbf{1}, -\dfrac{\alpha}{\boldsymbol{x}}\right)$ | $(\log \boldsymbol{y}, \boldsymbol{y})$ | $1$ | $\dfrac{\alpha \boldsymbol{y}}{\alpha - 1 - \boldsymbol{y}\odot \boldsymbol{z}}$ | $\dfrac{\alpha \boldsymbol{y}}{\alpha - 1 - \boldsymbol{y}\odot \boldsymbol{s}(\boldsymbol{y})}$ |
| Poisson | $\log(\lambda \boldsymbol{x})$ | $\lambda \boldsymbol{y}$ | $\dfrac{1}{\prod_{i=1}^{d}(\lambda y_i)!}$ | $\dfrac{1}{\lambda}\exp\left\{\dfrac{\boldsymbol{z}}{\lambda}\right\}$ | $\left(\boldsymbol{y}+\dfrac{1}{2\lambda}\right)\odot \exp\left\{\dfrac{\boldsymbol{s}(\boldsymbol{y})}{\lambda}\right\}$ |

## 1.13 The Proof of Eq. 17 in Section 3.2.3

*Proof.*

$$\nabla_{\boldsymbol{y}} \log p_{\mathbf{y}}(\boldsymbol{y}\mid\boldsymbol{x}) = \nabla_{\boldsymbol{y}} \log\left(p_{\mathbf{z}}(\bar{\boldsymbol{z}}\mid\boldsymbol{x})\left(1+\frac{\nabla_{\boldsymbol{z}}p_{\mathbf{z}}(\bar{\boldsymbol{z}}\mid\boldsymbol{x})^T(\boldsymbol{y}-\bar{\boldsymbol{z}})}{p_{\mathbf{z}}(\bar{\boldsymbol{z}}\mid\boldsymbol{x})}\right)\right)$$
$$\approx \nabla_{\boldsymbol{y}} \log p_{\mathbf{z}}(\bar{\boldsymbol{z}}\mid\boldsymbol{x}) + \nabla_{\boldsymbol{y}}\frac{\nabla_{\boldsymbol{z}}p_{\mathbf{z}}(\bar{\boldsymbol{z}}\mid\boldsymbol{x})^T(\boldsymbol{y}-\bar{\boldsymbol{z}})}{p_{\mathbf{z}}(\bar{\boldsymbol{z}}\mid\boldsymbol{x})}$$
$$= \frac{\nabla_{\boldsymbol{z}}p_{\mathbf{z}}(\bar{\boldsymbol{z}}\mid\boldsymbol{x})}{p_{\mathbf{z}}(\bar{\boldsymbol{z}}\mid\boldsymbol{x})} = \nabla_{\boldsymbol{z}} \log p_{\mathbf{z}}(\bar{\boldsymbol{z}}\mid\boldsymbol{x}).$$

$\square$

## 2   Conclusions of Gaussian, Gamma and Poisson Noise

Refer to Table 1.

## 3   Experiment

When training score function, for $\sigma_a$ in Eq. (29), we set initial value as $0.05$ and final value as $1\times 10^{-6}$. We reduce $\sigma_a$ linearly every 50 training steps and keep it as $1\times 10^{-6}$ for the final 50 steps. Another important point about the training for non-Gaussian noise model (from No.5 to No.10), we add a slight Gaussian noise to noisy images such that the score function estimation is stable and remove the additive Gaussian noise when inference as we do in mixture noise models. Here, we set the $\sigma$ of Gaussian noise as 5.

For Neighbor2Neighbor, We use the code in https://git-hub.com/TaoHuang2018/Neighbor2Neighbor and keep the default hyper-parameters setting.

For Noisier2Noise, we use the code in https://git-hub.com/melobron/Noisier2Noise. We set $\alpha = 1$ and compute the average of $50$ denoised results.