# OpenReview forum: "Unsupervised Image Denoising with Score Function"
_NeurIPS.cc/2023/Conference — NeurIPS 2023 poster_

### Official Review · Reviewer_Gx7h · 2023-06-18

**Soundness:** 3 good
**Presentation:** 4 excellent
**Contribution:** 2 fair
**Rating:** 5
**Confidence:** 3

**Summary:**

The paper introduces an extension of Noise2Score for unsupervised image denoising and demonstrates its efficiency on several noise models, including non-exponential family distributions.

**Strengths:**

The extension of Noise2Score allows to deal with different kinds of noise, as opposed to prior works that only focus on exponential family distributions. The paper is clearly-written, with a fair amount of examples and experiments.

**Weaknesses:**

The contribution is relatively incremental compared to Noise2Score, and is based on an approximation (equation (7) replaces equation (6) in the resolution) that is not really discussed. Even though Theorem 3.1 gives some upper bound on the distance between the estimate and the conditional expectation, it would have been interesting to get insights on the constant $C$ and on the covariance of $x$ given $y$ in practice, in particular with complex noise models the paper aims to deal with.

**Questions:**

* If possible, could the authors give numerical details on the approximation error between equations (6) and (7) in practical cases that differ from exponential family distributions, and elaborate on the hypotheses (f invertible, inverse function Lipschitz continuous, and bounded Hessian) ?

* Could the authors exhibit ill-posed cases for which this approximation wouldn't be accurate, and show the implication on the denoising algorithm ?

**Limitations:**

See remarks above.

---

> ### Author Rebuttal · Authors · 2023-08-08
>
> We thank the reviewer for the positive feedback and constructive comments. We answer the raised questions below. We hope that our answers clarify the doubts and address the concern of the reviewer.
>
> Q1: The contribution is relatively incremental compared to Noise2Score, and is based on an approximation (equation (7) replaces equation (6) in the resolution) that is not really discussed.
>
> A1: Our approach serves as an important expansion of Noise2Score. However it enables image denoising for multiple noise models (please refer to Table 1). Equation 7 is not an approximation of Equation 6; rather, it is derived based on the structure of Equation 6, akin to how Equation 5 is derived from Equation 4. We apologize if certain notations in the paper are causing confusion. To clarify, the "x" in Equations 5 and 7 should be replaced with $\hat{x}$. Equations 5 and 7 do not pertain to the original clean x and noisy y, but rather they concern an unknown denoised result, $\hat{x}$, with y representing the given and known noisy image. Consequently, both Noise2Score and our method aim to solve equations related to $\hat{x}$. However, Noise2Score arrives at the solving equation via Tweedie's Formula (the cases of exponential family distribution), while our method devises the solving equation based on Proposition 3.1 (more general distribution). In the majority of cases, including most exponential family distributions, the solution of Equation 7 does not yield E[x|y] (in other words, the approximation error between E[x|y] and the denoised result also exists in Noise2Score). As such, we employ Theorem 3.1 to gauge the closeness between the solution and E[x|y].
>
> Q2: Even though Theorem 3.1 gives some upper bound on the distance between the estimate and the conditional expectation, it would have been interesting to get insights on the constant and on the covariance of x given y in practice.
>
> A2: $C$ is related to the smoothness of score function, while the covariance of x given y is related to the noise level. A smoother score function and lower noise intensity, resulting in a smaller upper bound; conversely, a less smooth score function and higher noise intensity, leading to a larger upper bound.
>
> Q3: If possible, could the authors give numerical details on the approximation error between equations (6) and (7) in practical cases that differ from exponential family distributions, and elaborate on the hypotheses (f invertible, inverse function Lipschitz continuous, and bounded Hessian) ?
>
> A3: For an explanation regarding the approximation error, please refer to the response to the first question. Concerning hypotheses, please consult Proof 1.3 provided in the supplementary material. We list the explanations for them as follows:
>
> * f invertible: Given $\boldsymbol{y}$, $\boldsymbol{f}(\boldsymbol{x}, \boldsymbol{y})$ is invertible about $\boldsymbol{x}$.
> * inverse function Lipschitz continuous: Denote the inverse function of $\boldsymbol{f}(\boldsymbol{x}, \boldsymbol{y})$ as $\boldsymbol{f}_{\boldsymbol{y}}^{-1}$ where $\boldsymbol{y}$ is given. Thus the variable of the inverse function is $\boldsymbol{x}$. The Lipschitz continuous is an assumption on the property of $\boldsymbol{f}(\boldsymbol{x}, \boldsymbol{y})$.
> * bounded Hessian: Denote each component of $\boldsymbol{f}(\boldsymbol{x}, \boldsymbol{y})$ as $\boldsymbol{f}_i$, for any $i$, the Hessian matrix of $\boldsymbol{f}_i$ is bounded.

---

> > ### Comment · Reviewer_Gx7h · 2023-08-14
> > **Thanks for the answer**
> >
> > Thanks to the authors for their answer. After reading the other reviews and the rebuttal, I will keep my rating as is.

---

> > > ### Author Response · Authors · 2023-08-17
> > >
> > > Thank you for the comment!

---

### Official Review · Reviewer_Y7fg · 2023-07-04

**Soundness:** 3 good
**Presentation:** 3 good
**Contribution:** 2 fair
**Rating:** 5
**Confidence:** 4

**Summary:**

This paper presents a self-supervised learning algorithm for image denoising which only requires the score of the noisy image distribution to perform denoising. The algorithm builds on ideas of a recent paper Noise2Score, extending the family of noise distributions that can be handled via an application of Fisher's identity.


**Strengths:**

- The paper presents a principled way of performing self-supervised denoising with a large family of noise distributions, which go beyond the exponential family. The previous Noise2Score only handles distributions belonging to the exponential family.
- The paper demonstrates the good performance of the proposed approach on a large number of denoising experiments.

**Weaknesses:**

- The authors do not compare with SURE-based methods, which can handle mixture distributions, such as mixed Poisson Gaussian noise, e.g., see "An unbiased risk estimator for image denoising in the presence of mixed Poisson–Gaussian noise" by LeMontagner et al.

- While the idea is theoretically stimulating, I don't see a lot of practical applications for the family of distributions where SURE or Noise2Score would fail. It would be good if the authors could specify some.

- Some statements such as "Our approach is so powerful that" should be toned down. In particular, note that the approach cannot handle *any* noise distribution. See for example the discussion on Binomial noise in "Least squares estimation without priors or supervision" by Raphan and Simoncelli.

**Questions:**

If I'm not wrong, the main formula in Proposition 3.1 is just Fisher's identity? It would be good to name this accordingly and cite some references.

In Theorem 3.1, what does "f(x,y) is invertible" means? Invertible with respect to all variables? There is also a typo in Lipschitz continuous.

**Limitations:**

The limitations are not discussed in detail. It would be good to specify that the method strongly relies on the knowledge of the noise distribution, whereas other methods such as Neighbor2Neighbor do not.

---

> ### Author Rebuttal · Authors · 2023-08-08
>
> We thank the reviewer for the positive feedback and constructive comments. We answer the raised questions below. We hope that our answers clarify the doubts and address the concern of the reviewer.
>
> Q1: It would be good if the authors could specify some distributions where SURE or Noise2Score would fail.
>
> A1: The noise models of many real world data are not exponential family distribution or a mixture of exponential family distributions; for example, all the ultrasound images in both medical and non-medical applications have speckle noise (or approximated as Rayleigh noise), and thus our method would be more effective for these applications. More importantly, a lot of real world data has correlated noise (although the structure of correlation could be simple), such as the noise in medical images is always locally correlated because of partial volume effect due to limited resolution. When SURE or Noise2Score is applied to this type of application, it usually has to approximate the noise to uncorrelated noise.
>
> Q2: Is the main formula in Proposition 3.1 just Fisher's identity?
>
> A2: No, they are quite similar, but there are some differences.
>
> Fisher's identity:
> $$
> \nabla_\theta \log p_\theta\left(\boldsymbol{y}\right)=\int \nabla_\theta \log p_\theta\left(\boldsymbol{x}, \boldsymbol{y}\right) p_\theta\left(\boldsymbol{x} \mid \boldsymbol{y}\right) \mathrm{d} \boldsymbol{x}
> $$
> Prop 3.1:
> $$
> \nabla_{\boldsymbol{y}} \log p(\boldsymbol{y})=\int p(\boldsymbol{x} \mid \boldsymbol{y}) \nabla_{\boldsymbol{y}} \log p(\boldsymbol{y} \mid \boldsymbol{x}) \mathrm{d} \boldsymbol{x}
> $$
>
>
> Q3: In Theorem 3.1, what does "f(x,y) is invertible" means? Invertible with respect to all variables?
>
> A3: "f(x,y) is invertible" means that with y given, it is invertible with respect to x. This property make it possible to find a unique solution for x in Eq 7.

---

> > ### Comment · Reviewer_Y7fg · 2023-08-14
> >
> > Many thanks for answering my questions. After reading the rebuttal, I want to keep my original score.

---

> > > ### Author Response · Authors · 2023-08-17
> > >
> > > Thank you for the comment!

---

### Official Review · Reviewer_V5Ro · 2023-07-04

**Soundness:** 3 good
**Presentation:** 3 good
**Contribution:** 2 fair
**Rating:** 5
**Confidence:** 4

**Summary:**

The following work attempts to solve a single image denoising task in an unsupervised fashion. The authors propose to predict the score function and then denoise the input noisy image by solving the system of equations. Moreover, the proposed approach is a generalization of the Noise2Score method, which can work for complex non-exponential family distribution. Extensive experimental results demonstrate comparable and superior results in complex cases over existing unsupervised denoising methods.

**Strengths:**

+ Proposed extension of Noise2Score to images with complex noise distribution is very appealing and substantially increases the chances to apply the method to real-world scenarios.
+ Extensive experimental results on additive, multiplicative, and mixed noises.

**Weaknesses:**

- Proposed method is highly susceptible to noise parameter estimations according to Figure 2.
- Moreover, one needs to understand the noise distribution type (Gamma Noise, Poisson Noise, etc.) of an arbitrary input image to estimate particular parameters of the distribution. Then those estimations are further used to solve Eq. 7.
- It would be better to apply the proposed method to real image denoising given an estimated noise parameters. For example, raw image denoising (e.g. SIDD, DnD).


**Questions:**

Suggestions:
•	It would be better to move some of the details in the implementation part. For example, AR-DAE, which is used for score function estimation.
•	Typos (e.g. line 114)
•	I would suggest to add some visual examples in the Supplementary (if the main paper does not have enough space).

---

> ### Author Rebuttal · Authors · 2023-08-08
>
> We thank the reviewer for the positive feedback and constructive comments. We answer the raised questions below. We hope that our answers clarify the doubts and address the concern of the reviewer.
>
> Q1: Proposed method is highly susceptible to noise parameter estimations according to Figure 2.
>
> A1: Inaccurate estimation of noise parameters could lead to a decline in the performance of the proposed method. Nevertheless, a strategy akin to the one employed by Noise2Score for handling unknown noise parameters can be applied. This entails selecting an auxiliary indicator that operates independently of ground truth, such as Total Variation (TV). By solving Equation 7 with different noise parameters, a range of denoised outcomes is obtained. The denoised result exhibiting the most favorable performance according to the indicator is subsequently chosen.
>
> Q2: It would be better to apply the proposed method to real image denoising given an estimated noise parameters. For example, raw image denoising (e.g. SIDD, DnD)
>
> A2: We tested our approach on the SIDD dataset using the estimated parameters provided by the dataset, and produced a PSNR of 32.3. This result is better than the 32.2 achieved by unsupervised method Nr2N and comparable to the 33.9 achieved by supervised learning, demonstrating that our method is practical for real-world denoising problems.

---

> > ### Comment · Reviewer_V5Ro · 2023-08-21
> >
> > Dear authors, thank you for your detailed response. I hope that provided results on raw image denoising will be added to the manuscript, which definitely make the work stronger. As for the final assessment, I`m willing to keep my original score as is.

---

> > > ### Author Response · Authors · 2023-08-21
> > >
> > > Thank you for the comment! We tested our approach on the SIDD dataset using the estimated parameters provided by the dataset, and produced a PSNR of 32.3. This result is better than the 32.2 achieved by unsupervised method Nr2N and comparable to the 33.9 achieved by supervised learning, demonstrating that our method is practical for real-world denoising problems. We will provided more results on raw image denoising in the future.

---

### Official Review · Reviewer_PKR9 · 2023-07-06

**Soundness:** 3 good
**Presentation:** 3 good
**Contribution:** 3 good
**Rating:** 5
**Confidence:** 4

**Summary:**

This paper proposes a general unsupervised image denoising approach by solving system with an estimated score function. The proposed system can be applied on multiple noise models by changing the equation system rather than retraining the model.

**Strengths:**

1)	Compared to the Noise2Score limited to the noise models of exponential family distributions, the proposed method unlocks the limitation and generalizes to the non-exponential family distributions.
2)	For different noise model, the proposed method only requires to modify the equation system to be solved and keeps the trained score function estimator unchanged.
3)	The applications on the additive noise, multiplicative noise, and mixture noise have been proved in the manuscript and the supplementary materials.

**Weaknesses:**

1)	Qualitative comparison should be provided in the supplementary materials.
2)	The Noise2Score can be applied to the unknown noise parameters, and whether the proposed methods can achieve it? Please provide the discussion and experiments.

**Questions:**

The overall idea presented in this paper is interesting and the results are promising. Since I am not very sure about the application of the unknown noise parameters, I may defer my recommendation after authors' response.

**Limitations:**

N.A.

---

> ### Author Rebuttal · Authors · 2023-08-08
>
> We thank the reviewer for the positive feedback and constructive comments. We answer the raised questions below. We hope that our answers clarify the doubts and address the concern of the reviewer.
>
> Q1: Qualitative comparison should be provided in the supplementary materials.
>
> A1: We will offer a one-page PDF comprising a qualitative comparison. Afterward, we will incorporate it into the supplementary materials.
>
> Q2: The Noise2Score can be applied to the unknown noise parameters, and whether the proposed methods can achieve it?
>
> A2: Noise2Score tackles unknown noise parameters by adopting a strategy of selecting an additional indicator that operates without relying on ground truth, such as Total Variation (TV). This method involves solving Equation 7 for various noise parameters, resulting in diverse denoised outcomes. The denoised outcome displaying the most favorable indicator performance is subsequently selected. This approach is equally suitable for our method.

---

> ### Comment · Area_Chair_b54U · 2023-08-18
>
> Reviewer PKR9: Please respond to the rebuttal and give more details and context about your decision ASAP

---

### Official Review · Reviewer_Qfmm · 2023-07-23

**Soundness:** 4 excellent
**Presentation:** 3 good
**Contribution:** 3 good
**Rating:** 5
**Confidence:** 3

**Summary:**

This paper proposes a unsupervised image denoising method based on score function. Compared with Noise2Score, it discards the noise distribution on exponential family, and derives the solutions for additive Gaussion noise, multiplicative noise, and mixture noise. The experiments verify the effectiveness of the proposed method.

**Strengths:**

1. The  proposed method is theoretically sound.
2. It is a general unsupervised method, being able to deal with many kinds of noise types.
3. The synthetic experimental results showcase that it is superior or comparable to recent SoTA method.

**Weaknesses:**

1. The proposed method is built on the core equation of Eq. (7).  The detailed mathematical derivation of this equation is not clearly presented. One of the main contributions of this work is that it dose not rely on the exponential family distribution. However, according to my understanding, Eq. (7) still follows the exponential family distribution. The writing of this part should be further improved.

2. The introduction section states that "Another advantage of our approach is that regardless of noise models, the training process of the score function neural network is identical."  In my opinion, it still requires to individually train the models on various noise types.

3. The most limitation of this work is its practicality. It depends on the specific parameters of the noise distribution, thus cannot to deal with real-word denoising task. The experiments all focus on the synthetic datasets.

**Questions:**

See weakness.

**Limitations:**

As explained in the weakness, this work cannot handle the blind or real-world denoising task.

---

> ### Author Rebuttal · Authors · 2023-08-08
>
> We thank the reviewer for the positive feedback and constructive comments. We answer the raised questions below. We hope that our answers clarify the doubts and address the concern of the reviewer.
>
> Q1: The detailed mathematical derivation of this equation (Eq 7) is not clearly presented.
>
> A1: Equation 7 is derived based on the form of Equation 6, similar to how Equation 5 is derived based on the form of Equation 4. We apologize for any confusion stemming from certain notations in the paper. To clarify, the "x" in Equations 5 and 7 should be replaced with $\hat{x}$. Equations 5 and 7 are not equations about the clean x and noisy y, but rather about an unknown denoised result $\hat{x}$, where y is the given and known noisy image. Consequently, both Noise2Score and our approach aim to solve equations related to $\hat{x}$. However, Noise2Score derives its solving equation using Tweedie's Formula, while our method constructs the solving equation based on Proposition 3.1. As Equation 6 remains valid for any distribution, it naturally holds for exponential family distributions. If the distribution belongs to the exponential family, Equation 6 can be reformulated as Equation 4. Hence, under an exponential family distribution, Equation 7 can be expressed in the structure of Equation 5. In essence, our method (solving Equation 7) serves as an important expansion of Noise2Score that enables image denoising for multiple noise models.
>
> Q2: The introduction section states that "Another advantage of our approach is that regardless of noise models, the training process of the score function neural network is identical." In my opinion, it still requires to individually train the models on various noise types.
>
> A2: The underlying idea here is that when the noise model is provided, the training approach remains unchanged irrespective of its type and parameters. This approach focuses on estimating the score function, and the estimation method is independent of the noise model itself. In real-world scenarios, it's necessary to estimate the noise model's type and parameters in advance. If any changes occur in these estimations, the original score function estimation remains applicable as the noisy image remains unchanged. Consequently, retraining the model isn't needed; modifying Equation 7 and solving it anew would be sufficient.
>
> Q3: The most limitation of this work is its practicality. It depends on the specific parameters of the noise distribution, thus cannot to deal with real-word denoising task.
>
> A3: This is a limitation of our method, but it can be addressed by two approaches. For real-world denoising tasks: 1) Employ domain knowledge to estimate noise parameters; 2) Follow a method mentioned in Noise2Score, selecting an additional indicator that doesn't rely on ground truth, such as Total Variation (TV). By solving Equation 7 for various noise parameters, distinct denoised outcomes are achieved. The most optimal denoised outcome is then selected based on the indicator's performance. In fact, Noise2Score also has this limitation and Noise2Score addresses it by the second approach.

---

> > ### Comment · Reviewer_Qfmm · 2023-08-14
> > **Response to authors**
> >
> > 1. I suggest the authors to correct the notation mistakes of Eq.(5) and Eq. (7) in the revised version.
> >
> > 2. Reviewer PKR9 and Reviewer V5Ro also concerns the application of the proposed method in real-world denoising dataset. If some quantitative comparison results can be provided, I tend to increase my rating.

---

> > > ### Author Response · Authors · 2023-08-17
> > >
> > > Thank you for the comment! We tested our approach on the SIDD dataset using the estimated parameters provided by the dataset, and produced a PSNR of 32.3. This result is better than the 32.2 achieved by unsupervised method Nr2N and comparable to the 33.9 achieved by supervised learning, demonstrating that our method is practical for real-world denoising problems.

---

### Author Rebuttal · Authors · 2023-08-09

We thank the reviewer for the positive feedback and constructive comments. The qualitative comparison is in the attached PDF.

---

### Decision · Program_Chairs · 2023-09-21

**Decision:**

Accept (poster)

**Comment:**

Overall, after the rebuttal round, the reviews are positive about the paper and all recommend it for (borderline) acceptance. I tend to agree that the paper has made a worth contribution and that the authors have made a convincing effort to address the shortcomings pointed out by the reviewers. Some lingering concerns remain about the mathematical derivation of Eq. 7 and the practicality of the proposed method, but these concerns are not sufficient to keep the paper from being accepted. That said, it would be very nice if the authors could, at least in future work, conduct additional experiments to demonstrate the practicality of the proposed method on real-world datasets.